# REVISITING TEXT-TO-IMAGE EVALUATION WITH GECKO: ON METRICS, PROMPTS AND HUMAN RATING

**Olivia Wiles**[*,†]    **Chuhan Zhang**[*,†]    **Isabela Albuquerque**[*,†]    **Ivana Kajić**[†]    **Su Wang**[†]

**Emanuele Bugliarello**[†]    **Yasumasa Onoe**[†]    **Pinelopi Papalampidi**[†]    **Ira Ktena**[†]

**Chris Knutsen**[†]    **Cyrus Rashtchian**[‡]    **Anant Nawalgaria**[§]    **Jordi Pont-Tuset**[†]

**Aida Nematzadeh**[†]

## ABSTRACT

While text-to-image (T2I) generative models have become ubiquitous, they do not necessarily generate images that align with a given prompt. While many metrics and benchmarks have been proposed to evaluate T2I models for alignment, the impact of the evaluation components (prompt sets, human annotations, evaluation task) has not been systematically measured. We find that looking at only *one slice of data*, i.e. one set of skills or human annotations, is not enough to obtain stable conclusions that generalise to new conditions or slices when evaluating T2I models or alignment metrics. We address this by introducing an evaluation suite of >100K annotations across four human annotation templates that comprehensively evaluates models' capabilities across a a range of methods for gathering human annotations and comparing models. In particular, we propose (1) a carefully curated set of prompts – *Gecko2K*; (2) a statistically grounded method of comparing T2I models; and (3) a framework to systematically evaluate metrics under three *evaluation tasks – model ordering, pair-wise instance scoring, point-wise instance scoring*. Using this evaluation suite, we compare a wide range of metrics and find that a given metric may do better in one setting but worse in another. As a result, we introduce a new, interpretable auto-eval metric that is consistently better correlated with human ratings than existing ones on our evaluation suite–across different human templates and evaluation settings–and on TIFA160.

## 1    INTRODUCTION

Text-to-image (T2I) models (Saharia et al., 2022; Yu et al., 2022b; Betker et al., 2023; Rombach et al., 2022) generate images of impressive quality, but the images are not necessarily aligned with the prompt. The key to comparing T2I models is in the dataset of prompts and human annotations we collect. Human annotation is slow and expensive, motivating the creation (Hu et al., 2023; Cho et al., 2023a) of automatic-evaluation (auto-eval) metrics as a replacement. To evaluate both metrics and models, human annotation is the gold standard. However, Clark et al. (2021) show that the template design and annotator knowledge can significantly impact results in the text domain. In this work, we create a comprehensive benchmark to answer the question: *how do the choices around prompts and human annotation templates impact our metric and modelling decisions?*

There has been limited work analysing the impact on model and metric ranking due to these choices. Previous work builds a benchmark by collecting annotations across *one* template and prompts that cover a limited distribution of skills (see Table 1 for a comparison). A skill refers to a generation

---

[*]Equal contribution. Correspondence to: oawiles@google.com; nematzadeh@google.com.
[†]Google DeepMind, [‡]Google Research, [§]Google Cloud.
Github link: https://github.com/google-deepmind/gecko_benchmark_t2i

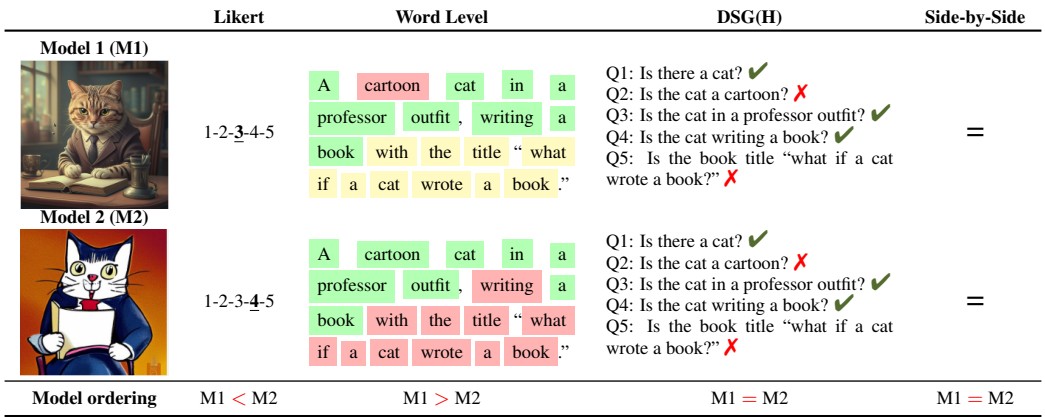

Figure 1: **Model ordering outcomes for one annotation template do not necessarily generalise to other templates.** We generate images for two models using the prompt: *A cartoon cat in a professor outfit, writing a book with the title "what if a cat wrote a book."*. By collecting extensive human evaluation, we expose disparities across templates: outcomes between T2I models or auto-eval metric obtained for one template may not generalise to others.

challenge, such as text rendering or generating different colors and shapes and a sub-skill refers to sub-challenges (e.g. generating longer text or Gibberish). Other work does not systematically gather prompts to ensure a wide coverage of skills and properties with the exception of Zhu et al. (2023). Cho et al. (2023a) does consider different lengths of prompts and Zhu et al. (2023) some skills but this is done for a specific template and does not consider varied challenges for a given skill.

By looking at *one slice of data* (e.g., too few or too specific prompts or one human annotation template), we are at risk of drawing conclusions that are *specific to that slice and do not generalise*. As a result, we collect a comprehensive dataset (Table 1) systematically over different prompt sources and different skills/subskills. Given this prompt set, we generate images from four T2I models and rate them across four human annotation templates. By considering model rankings across annotation templates for a given prompt, we can further determine how reliably a prompt measures alignment.

Similarly, the choice of task may impact our results. Auto-eval metrics are typically evaluated using correlation with human judgement. However, in practice, we would want to use metrics for three tasks: (1) model ordering, ranking T2I models based on *significant* relationships; (2) pair-wise instance scoring, choosing whether a given output is better than another and (3) point-wise instance scoring, an estimation of a samples' overall alignment. Evaluating metrics on one task is not enough: we may *think* we are choosing the best metric for all three but we show that conclusions for one task *do not necessarily generalise*. An overview of our contributions follows:

- *Gecko*: An evaluation suite for T2I alignment which includes a comprehensive set of 2K prompts, 4 human templates to evaluate 4 T2I models to give ∼100K human annotations (Table 1). We get predictions from a wide range of auto-eval metrics and evaluate under 3 realistic settings (model ordering, pair-wise instance scoring, point-wise instance scoring).

- Using our suite, we demonstrate limitations of looking at a single slice of data as currently done in the literature: different metrics and models show different results depending on the prompt slice or template.

- Based on our analyses, we introduce an interpretable state-of-the-art QA/VQA metric. It gets the most number of model comparisons right, and performs on average 40.5%/22% better than interpretable baselines on our dataset in terms of pair-wise instance scoring and point-wise instance scoring respectively, and 10.5% better on TIFA160 (Hu et al., 2023).

## 2 RELATED WORK

**Benchmarking alignment in T2I models.** Many benchmarks have been proposed to holistically evaluate model capabilities within T2I alignment. Early benchmarks are small scale and created

| | Likert | Word Level | DSG(H) | SxS | # prompts annotated | #anns/#img | # anns | # Skills (All Categories) | # Sub-Skills |
|---|---|---|---|---|---|---|---|---|---|
| DSG1K (Cho et al., 2023a) | ✗ | ✗ | ✓ | ✗ | 1.06K | 3 | 9.6K | 11(13) | ✗ |
| DrawBench (Saharia et al., 2022) | ✗ | ✗ | ✗ | ✓ | 200 | 25 | 25K | 7(11) | ✗ |
| PartiP. (Yu et al., 2022b) | ✗ | ✗ | ✗ | ✓ | 1.6K | 5 | 16K | 9(23) | ✗ |
| TIFA160 (Hu et al., 2023) | ✓ | ✗ | ✗ | ✗ | 160 | 2 | 1.6K | 8(12) | ✗ |
| PaintSkills (Cho et al., 2023b) | ✓ | ✗ | ✗ | ✗ | 150 | 5 | 2.25K | 3(3) | ✗ |
| W-T2I (Zhu et al., 2023) | ✓ | ✗ | ✗ | ✗ | 200 | 3 | 2.4K | 15(20) | 9 |
| HEIM (Lee et al., 2024) | ✓ | ✗ | ✗ | ✗ | 708 | ∼5.4 | ∼150K | 6(6) | ✗ |
| Gecko2K | ✓ | ✓ | ✓ | ✓ | **2K** | ∼13.5 | ∼108K | 12(12) | 36 |
|   Gecko(R) | ✓ | ✓ | ✓ | ✓ | 1K | ∼13.5 | ∼54K | 11(11) | ✗ |
|   Gecko(S) | ✓ | ✓ | ✓ | ✓ | 1K | ∼13.5 | ∼54K | 12(12) | 36 |

Table 1: **Comparison of annotated alignment datasets.** We report the amount of human annotation and skill division for each dataset. We can see that many datasets include only a handful of annotated prompts or a small number of annotations (anns) per image or overall. No dataset besides Gecko2K collects ratings across multiple different human annotation templates. We also include the number of skills and sub-skills in each dataset. Again, Gecko includes the most number of sub-skills, allowing for a fine-grained evaluation of metrics and models. When datasets do not include skills, we map their categories into skills/sub-skills as appropriate.

alongside model development to perform side-by-side model comparisons (Saharia et al., 2022; Yu et al., 2022b; Betker et al., 2023). Later work (e.g., TIFA (Hu et al., 2023), DSG1K (Cho et al., 2023a) and HEIM (Lee et al., 2024)) focuses on creating holistic benchmarks by drawing from existing datasets (e.g., MSCOCO (Lin et al., 2014), Localized Narratives (Pont-Tuset et al., 2020) and CountBench (Paiss et al., 2023)) to evaluate a range of capabilities including counting, spatial relationships, and robustness. Other datasets focus on a specific challenge such as compositionality (Huang et al., 2024a), contrastive reasoning (Zhu et al., 2023), text rendering (Tuo et al., 2023), reasoning (Cho et al., 2023b), spatial reasoning (Gokhale et al., 2022), or specifically image ordering given an increasing number of errors (Saxon et al., 2024). The Gecko2K benchmark is similar in spirit to TIFA and DSG1K in that it evaluates a set of skills. However, in addition to drawing from previous datasets—which may be biased or poorly representative of the challenges of a particular skill—we collate prompts across sub-skills for each skill to obtain a discriminative prompt set. Moreover, we gather human annotations across multiple templates and many prompts (see Table 1).

**Automatic metrics measuring T2I alignment.** Inspired by work in image captioning, a widely used auto-eval metric is CLIPScore (Hessel et al., 2021). However, such metrics poorly capture finer-grained aspects of images (Bugliarello et al., 2023; Yuksekgonul et al., 2022). Motivated by work in NLP on evaluation using entailment or QA metrics (Maynez et al., 2020; Kryściński et al., 2019; Honovich et al., 2021), similar metrics (Yarom et al., 2024) have been devised for T2I alignment. However, such a metric may not generalise to new settings and is not interpretable—one cannot diagnose why an alignment score is given. Visual question answering (VQA) methods such as TIFA (Hu et al., 2023), $VQ^2$ Yarom et al. (2024) and DSG (Cho et al., 2023a) do not require task-specific finetuning and give an interpretable explanation for their score. These metrics create QA pairs which are then scored with a VLM given an image and aggregated into a single score. However, the performance of such methods is conditional on the behaviour of the underlying LLMs used for question generation, and VLMs used for answering questions.

## 3   *Gecko2K:* THE GECKO BENCHMARK

We curate a fine-grained skill-based benchmark, Gecko2K, with good coverage by curating two sets of prompts: one created systematically based on a set of skills and subskills (Gecko(S)) and one generated by combining existing datasets but tagging them and resampling to ensure good coverage over those tags (Gecko(R)). We generate Gecko(R) by extending the DSG1K (Cho et al., 2023a) benchmark creation approach to use automatic tagging and improve the distribution of skills and linguistic properties (see App. B for details on the automatic tagging). However, due to the automatic tagging and nature of the underlying datasets, Gecko(R) is limited in the skills/sub-skills it covers. To generate our systematic set, we propose a hierarchical method combined with LLM generation in order to ensure a systematic distribution across skills (e.g., *counting*) and subskills (e.g., *simple modifier*: '1 cat' vs *additive*: '1 cat and 3 dogs'). This notion of sub-skills ensures we are capturing a wide distribution of prompts and not just one easy slice (e.g.generating counts of 1-4 objects).

## 3.1 GECKO(R): RESAMPLING DAVIDSONIAN SCENE GRAPH BENCHMARK

The recent DSG1K benchmark (Cho et al., 2023a) curates a list of prompts from existing image-text datasets[*] but does not control for the coverage or complexity of a given skill. The authors randomly sample 100 prompts and limit the prompt length to 200 characters. The resulting dataset is imbalanced in terms of the distribution of skills. Also, as T2I models take in longer and longer prompts, the dataset will not test models on that capability. We take a principled approach in creating Gecko(R) by resampling from the base datasets in DSG1K for better coverage and lifting the length limit. After this process, there are 175 prompts longer than 200 characters and a maximum length of 570 characters. Also, this new dataset has better coverage over a variety of skills than the original DSG1K dataset (see Fig. 6).

While resampling improves the distribution of skills, Gecko(R) has the following shortcomings. Due to the limitations of automatic tagging, it does not include all skills we wish to explore (e.g., language). It also does not include sub-skills: e.g., text rendering prompts do not focus on numerical text, or longer text (see Fig. 7). Finally, automatic tagging can be error prone.

## 3.2 GECKO(S): A CONTROLLED AND DIAGNOSTIC PROMPT SET

The aim of Gecko(S) is to generate prompts in a controllable manner for skills that are not well represented in previous work. We divide skills into sub-skills to diversify the difficulty and content of prompts. We take inspiration from psychology literature where possible (e.g., colour perception) and known limitations of current T2I models.

**Curating a controlled set of prompts with an LLM.** To generate a set of prompts semi-automatically, we use an LLM. We first decide on the sub-skills we wish to test for. For example, for text rendering, we may want to test for (1) English vs Gibberish to evaluate the model's ability to generate uncommon words, and (2) the length of the text to be generated. We then create a template which conditions the generation on these properties. Note that as we can

| Sub-skill | Example |
|---|---|
| Numbers / symbols | equation of "3+4 = 7" etched into a rock |
| Length | a neon sign with the words "the future is already here..." reflected on a rainy street. (n=29) |
| Gibberish | graffiti made with bright pink paint on the concrete, saying "fluff floop floof!" |
| Typography | "i love you" written in serif font in grass |

Table 2: **Subcategories and corresponding motivations for the text rendering skill.**

generate as much data as desired, we can define a distribution over the properties and control the number of examples generated for each sub-skill. Finally, we run the LLM and manually validate that the prompts are reasonable, fluent, and match the conditioning variables (e.g., the prompt has the right length and is Gibberish / English). A sample template is given in App. B.3.

**Gecko(S) make up.** Using this approach and also some manual curation, we focus on twelve skills falling into five categories (Fig. 5): (1) NAMED ENTITIES; (2) TEXT RENDERING; (3) LANGUAGE/LINGUISTIC COMPLEXITY; (4) RELATIONAL: ACTION, SPATIAL, SCALE; (5) ATTRIBUTES: COLOR, COUNT, SURFACES (TEXTURE/MATERIAL), SHAPE, STYLE. For sub-skills, we give a full breakdown for all skills in App. B.4. In Table 2 we give examples of the sub-skills and corresponding prompts for TEXT RENDERING. Using this approach, we get better coverage over the given sub-skills than other datasets (including Gecko(R)) as shown in Fig. 7.

## 4 COMPARING ANNOTATION TEMPLATES FOR MODELLING

We examine how the choice of human annotation template impacts results when comparing four models: SD1.5 (Rombach et al., 2022), SDXL (Podell et al., 2023), Muse[†] (Chang et al., 2023), and Imagen Vermeer (Vasconcelos et al., 2024). We consider *absolute comparison* templates (i.e., Likert, Word Level from Liang et al. (2023), and DSG(H) from Cho et al. (2023a)) which evaluate models individually, and a template for *relative comparison* of two models (side-by-side or SxS). A

---

[*]TIFA(Hu et al., 2023), Stanford Par.(Krause et al., 2017), Localized Narr.(Pont-Tuset et al., 2020), Count-Bench (Paiss et al., 2023), VRD (Lu et al., 2016), DiffusionDB (Wang et al., 2022), MJ (Turc & Nemade, 2023), PoseScript (Delmas et al., 2022), Whoops (Bitton-Guetta et al., 2023), DrawText-Creative (Liu et al., 2022).

[†]Muse is based on the original model, but trained on different data sources.

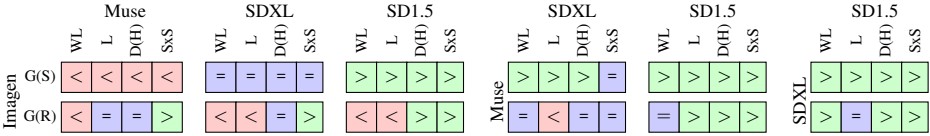

Figure 2: **Comparing models using human annotations.** We compare model rankings on Gecko(S)/(R). Each grid represents a comparison between two models. Entries in the grid depict results for WL, Likert (L), DSG(H) (D(H)), and side-by-side (SxS) scores. The > sign indicates the left-side model is better, worse (<), or not significantly different (=) than the model on the top.

high-level visualisation of each template is in Fig. 1 and details in App. D.1. We further introduce a principled method to determine significant model orderings based on human judgements.

## 4.1 Data Quality

We validate the reliability of each template and examine if the choice of the template impacts the quality of the collected data. Given the collected human ratings across the three templates, we compute inter-annotator agreement (IAA) for each generative model by measuring Krippendorff's $\alpha$, $\mathcal{K}_\alpha \in [-1, 1]$ (Hayes & Krippendorff, 2007), where a value of 1 indicates perfect agreement and 0 chance (Zapf et al., 2016). Results reported in Table 3 show that agreements are all high, with $\alpha > 0.5$, except for the Likert—SD1.5 pair for Gecko(R); we conjecture this is due to the lower quality images of SD1.5. Overall we find that fine-grained templates (WL and DSG(H)) are more reliable (e.g., have higher IAA) and WL achieves the highest IAA for the diverse Gecko(R). We also measure IAA for the SxS template in Table 11. We see lower IAA for the SxS template (though still far above chance) compared to the fine-grained ones. The IAA is < 0.5 for 6 out of 12 model comparisons. It seems, given the same number of annotators, the SxS template is less reliable.

**Reliable prompts.** Upon manual investigation, we find that differences in human ratings across templates can arise when prompts are difficult to judge with respect to alignment (and not due to the choice of the template): for example, when a prompt contains domain specific knowledge such as *"A bottle of Irn-Bru is sitting on a shelf"* or subjective notions such as *"a futuristic sculpture"*. To understand how this impacts our results, we consider a subset of the prompts that achieve high IAA across templates and models. For each model and absolute template, we select the prompts for which inter-rater *disagreement*[‡] is < 50% of the maximum *disagreement* observed across all prompts for that model–template pair. The intersection of these prompts across models and templates gives our *reliable prompts*. We additionally remove instances where all Likert ratings are *Unsure* to get 531 and 725 *reliable prompts* for Gecko(R) and Gecko(S), respectively. We first validate that using reliable prompts increases IAA on the SxS template (which was not used in the selection process) and find that it increases the average $\mathcal{K}_\alpha$ from 0.45 to 0.47 on Gecko(R), and 0.49 to 0.54 on Gecko(S) (see App. D.2 for details). In the next sections, we demonstrate how this subset of prompts increases agreement among templates, but at the expense of removing some potentially meaningful prompts.

## 4.2 Absolute Annotation Templates: Comparing T2I Models

**Average ratings.** For the absolute annotation templates, previous work compares T2I models by comparing the average ratings across examples. We report these values in Table 3 and find that the chosen prompt set impacts which model is best (e.g. SDXL for Gecko(R) and Muse for Gecko(S)). Moreover, the model with the lowest rating depends on both the prompt set and the template: given Gecko(R), Imagen is worse if using Likert, but SD1.5 is worse if using DSG(H). This highlights the importance of examining models in various conditions. When using T2I models in practice, we need to make conclusions about model ordering with high confidence. We argue that this evaluation is not enough: it does not measure if the difference between models is significant, which is particularly important as models start to saturate. As a result, we introduce the model ordering task.

**Model ordering.** We verify the significance of outcomes by performing the Wilcoxon signed-rank test with $p < 0.001$. Where results indicate the null-hypothesis is rejected (i.e., the distribution

---

[‡]Defined as the variance across image, word, and question ratings for Likert, WL and DSG(H) respectively.

| Gen. model | Inter annotator agreement | | | | | | Scores | | | | | |
|---|---|---|---|---|---|---|---|---|---|---|---|---|
| | Gecko(R) | | | Gecko(S) | | | Gecko(R) | | | Gecko(S) | | |
| | WL | Likert | DSG(H) | WL | Likert | DSG(H) | WL | Likert | DSG(H) | WL | Likert | DSG(H) |
| Imagen | **0.81** | 0.64 | 0.68 | 0.72 | 0.57 | **0.75** | $0.74_{\pm0.30}$ | $0.60_{\pm0.22}$ | $0.84_{\pm0.18}$ | $0.80_{\pm0.24}$ | $0.59_{\pm0.20}$ | $0.78_{\pm0.23}$ |
| Muse | **0.82** | 0.78 | 0.72 | 0.69 | 0.58 | **0.72** | $0.84_{\pm0.24}$ | $0.61_{\pm0.25}$ | $0.83_{\pm0.22}$ | $\mathbf{0.88}_{\pm0.18}$ | $\mathbf{0.63}_{\pm0.21}$ | $\mathbf{0.84}_{\pm0.21}$ |
| SDXL | 0.75 | **0.76** | 0.57 | 0.67 | 0.56 | **0.70** | $\mathbf{0.87}_{\pm0.19}$ | $\mathbf{0.68}_{\pm0.22}$ | $\mathbf{0.86}_{\pm0.16}$ | $0.80_{\pm0.23}$ | $0.60_{\pm0.21}$ | $0.79_{\pm0.22}$ |
| SD1.5 | **0.66** | 0.36 | **0.66** | 0.69 | 0.59 | **0.74** | $0.86_{\pm0.16}$ | $0.67_{\pm0.22}$ | $0.76_{\pm0.23}$ | $0.61_{\pm0.33}$ | $0.49_{\pm0.21}$ | $0.68_{\pm0.27}$ |

Table 3: **Inter-annotator agreement and ratings for all models and templates.** We measure inter-annotator agreement for each human evaluation template with Krippendorff's $\alpha$. Higher values indicate better agreement. We also show the mean and std. deviation for the annotated judgements of all templates after mapping the ratings to the $[0, 1]$ interval, with 1 indicating perfect alignment.

of ratings is significantly different), we can say that one model is better than another. To determine which model is best, we compare the mean values of their ratings. In Fig. 2 we visualise the outcomes for all model pairs across all templates. We see that Muse is not worse than any of the contenders across all templates and prompt sets, except for 2 out of the 12 comparisons involving Muse for Gecko(R); we determine it is the best overall model. In contrast with the results presented in Table 3, where SDXL is identified as the best model for Gecko(R) across all the templates, we observe that the significance results reveal that Muse and SDXL actually have similar performance, showcasing the importance of determining significance before drawing conclusions.

**Reliable prompts.** Constraining Gecko(R) using the reliable subset decreases the number of conflicts between the different templates, but at the potential expense of comparing models on fewer, potentially easier, prompts. Considering the two prompt sets, we observe that when using the synthetic prompts, Gecko(S)-rel, all templates agree in Fig. 13 in Appendix D.2. We hypothesise this is because the skills (e.g., color or shape), while hard to generate, are easy to evaluate within generation. For Gecko(R)-rel, we see disagreements between templates, where surprisingly, DSG(H) often result in a different relation than the two other templates. We also consider the full prompt-set, Gecko2K-rel, as it better captures the overall use cases of T2I models: we find that there is always a majority agreement, and the two fine-grained templates (WL and DSG(H)) always agree.

**Results by skill.** We explore how human judgements vary by skill and template; average ratings for each absolute template are shown in Fig. 23-Fig. 26 in the appendix. A lower average ratings per skill across templates indicates how 'challenging' a given skill is: we can see that 'lang compositional', 'lang complexity', 'count' and 'text' are consistently difficult across templates.

### 4.3 RELATIVE ANNOTATION TEMPLATE: COMPARING T2I MODELS

**Model ordering.** For the SxS template, a model is considered better if it is chosen as preferred more often than the competitor and the *Unsure* rating. To assess statistical significance, we perform a similar procedure as for the absolute annotation templates using binary scores for the ratings: 0 when there was a tie, +1 when a model was preferred by the majority of raters, and -1 otherwise.

We also compare the SxS template with the considered absolute annotation templates by computing the accuracy obtained by each absolute template when predicting the preferred model given by SxS on Gecko2K-rel. Results presented in Table 12 in App.D.2 show that all absolute annotation templates predict SxS judgements with similar average accuracy of around 70%, with DSG being the overall best. This shows that, although the results of pairwise model comparisons are the same in many cases for Gecko2K-rel as shown in Fig. 13, absolute and side-by-side annotations do not necessarily correspond to the same model ordering at the datapoint level.

> **Takeaway 1:** Fine-grained templates (i.e. ones that require multiple annotations per example), WL and DSG(H), yield the highest inter-annotator agreement. **Takeaway 2:** All three absolute annotation templates achieve similar, but not perfect, accuracy when predicting relative comparison annotations for each datapoint. **Takeaway 3:** To compare models reliably, we need to measure the *significant model ordering*. Model ordering depends on the human template and prompt set, but some prompt sets lead to consistent agreement across templates (e.g., are *discriminative*) such as our skill-based Gecko(S) or the larger, reliable set Gecko2k-Rel.

| Metrics | Zero-shot | Gecko(R) | | | | Gecko(S) | | | |
|---|---|---|---|---|---|---|---|---|---|
| | | WL | Likert | DSG(H) | SxS | WL | Likert | DSG(H) | SxS |
| | | | SpearmanR | | Acc | | SpearmanR | | Acc |
| *Interpretable (QA/VQA)* | | | | | | | | | |
| TIFA$_{\text{PALM-2/PALI}}$ | ✓ | 0.26 | 0.34 | 0.28 | 41.7 | 0.39 | 0.32 | 0.39 | 53.2 |
| DSG$_{\text{PALM-2/PALI}}$ | ✓ | 0.35 | 0.47 | 0.42 | 49.6 | 0.45 | 0.45 | 0.45 | 58.1 |
| Gecko$_{\text{PALM-2/PALI}}$ | ✓ | 0.41 | 0.55 | 0.46 | 62.1 | 0.47 | 0.52 | 0.45 | 74.6 |
| Gecko$_{\text{Gemini Flash}}$ | ✓ | **0.43** | **0.58** | **0.48** | 72.2 | **0.54** | **0.59** | **0.56** | 78.8 |
| *Uninterpretable (single score)* | | | | | | | | | |
| CLIP | ✓ | 0.14 | 0.16 | 0.13 | 54.4 | 0.25 | 0.18 | 0.26 | 67.2 |
| PyramidCLIP | ✓ | 0.26 | 0.27 | 0.26 | 64.3 | 0.22 | 0.25 | 0.23 | 70.7 |
| VQAScore$_{\text{Gemini Flash}}$ | ✓ | 0.42 | 0.54 | 0.45 | 73.1 | 0.51 | 0.57 | 0.49 | 76.5 |
| VNLI | ✗ | 0.37 | 0.49 | 0.42 | 54.4 | 0.45 | 0.55 | 0.45 | 72.7 |

Table 4: **Correlation between auto-eval metrics and human ratings across annotation templates on Gecko2K.** With the same backend, Gecko outperforms all other QA/VQA metrics across all evaluations and Gecko with GeminiFlash performs even better; it performs better or similar to the strongest single-score approach (VQAScore). **Bold**: Top results. Underlined: Top results by category.

## 5 THE GECKO METRIC

An auto-eval metric is more useful if it is (1) interpretable—it reports where a model fails in addition to its overall goodness, (2) reference-free–does not require a reference distribution, and (3) modular—can easily leverage better pretrained models for improved performance. As a result, we focus on improving recent work using a two-stage QA/VQA metric (Hu et al., 2023; Cho et al., 2023a; Yarom et al., 2024) that matches this criteria (as opposed to metrics such as VNLI (Yarom et al., 2024) and CLIP (Radford et al., 2021)). However, the QA/VQA pipelines are impacted by the shortcomings of the pretrained models used. In particular, we identify two main limitations of these pipelines and address them: the QA generation is not always *grounded* in the prompt as the generated questions might not necessarily cover *all* key parts of the prompt and also there might be *hallucinated* questions that are not related to the prompt. Moreover, at the VQA stage, the highest scoring answer might still be low probably but is treated as the "right" answer—we model this *uncertainty* in the VQA responses. Finally, we simplify the previously proposed methods by removing complexities (such as scene graph generation in DSG) and show that our simplified and improved setup is significantly better across the board.

A standard QA setup (e.g., Hu et al. (2023)) consists of three steps: (1) QA generation: prompting an LLM to generate a set of binary question-answer pairs $\{Q_i, A_i\}_{i=1}^N$ on a given T2I text description $T$. (2) VQA assessment: employing a VQA model to predict answer $\{A'_i\}_{i=1}^N$ for the generated questions given the generated image $I$. (3) Scoring: computing the alignment score by assessing the VQA accuracy using Eq. (1):

$$Alignment(T, I) = \frac{1}{N} \sum_{i=1}^N \mathbb{1}[A'_i = A_i]. \tag{1}$$

**Groundedness: increasing coverage.** To ensure the coverage of questions over the key elements in a text sentence $T$, we split the QA generation into two steps. We first prompt the LLM to index the visually groundable words in the sentence. For example, the sentence "*A red colored dog.*" is transformed into "*A {1}[red colored] {2}[dog].*" Subsequently, using the text with annotated keywords $\{W'_i\}_{i=1}^N$ as input, we prompt the LLM again to generate a QA pair $\{q_i, a_i\}$ for each word labelled $\{w'_i\}$ in an iterative manner (see App. C for the prompting details). This two-step process ensures a more comprehensive and controllable QA generation process, particularly for complex or detailed text descriptions where the prompted LLM often selectively generates questions for specific segments of the text while overlooking others.

**Groundedness: removing hallucination.** LLMs can hallucinate (Bang et al., 2023; Guerreiro et al., 2023), leading to the generation of low-quality, unreliable QA pairs. We filter out hallucinated QA pairs by taking inspiration from previous work in NLP (Maynez et al., 2020; Kryściński et al., 2019): we employ a Natural Language Inference (NLI) model (Honovich et al., 2022) model for measuring the factual consistency between the text $T$ and QA pairs $\{Q_i, A_i\}$. QA pairs with a consistency score lower than a threshold $r$ are removed, ensuring that the remaining QAs are about the prompt.

**Uncertainty: VQA score normalisation.** Finally, we improve aggregation of scores from the VQA model. The reliance on binary judgement—strictly matching $A'_i$ and $A_i$ without considering the

| Metrics | Gecko(R) | | | Gecko(S) | | |
|---|---|---|---|---|---|---|
| | WL | Likert | DSG(H) | WL | Likert | DSG(H) |
| | Pearson | | | | | |
| TIFA baseline | 0.21 | 0.32 | 0.25 | 0.39 | 0.32 | 0.39 |
| + coverage | 0.28 | 0.34 | 0.32 | 0.41 | 0.33 | 0.40 |
| + VQA score norm | 0.32 | 0.42 | 0.37 | 0.43 | 0.37 | 0.41 |
| + NLI filtering | **0.38** | **0.51** | **0.42** | **0.46** | **0.48** | **0.46** |

Table 5: **Validation of each component of the proposed Gecko metric on Gecko2K.** We evaluate the utility of the three proposed improvements by adding them to the TIFA baseline one by one. They all bring higher correlation with human judgement across the board on Gecko2K.

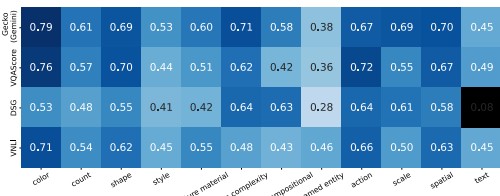

Figure 3: **Per skill results of different metrics.** Likert correlation for each skill; square black indicates p-values $> 0.05$. Full results in App. F.

predicted probability of $A'_i$—overlooks the inherent uncertainty in the predictions; a VQA model can predict a very similar score for two answers. If we simply take the max, then we lose this notion of uncertainty reflected in the scores. As a result, we normalise the scores as follows,

$$Alignment(T, I) = \frac{1}{N} \sum_{i=1}^{N} \frac{s_a}{\sum_i s_i}, \qquad (2)$$

where the negative log likelihood of answer $A'_i$ is $s_i$ and the correct answer is $A'_a$ with score $s_a$.

## 6 EXPERIMENTS ON AUTO-EVAL METRICS

We evaluate metrics across multiple prompt sets and templates to determine how they fare on the three tasks: (1) Do they give a good numeric measurement of overall alignment – **point-wise instance scoring**; (2) Are they good indicators on side by side comparisons – **pair-wise instance scoring**; (3) Can they predict **model ordering**. We demonstrate that the task *can* impact rankings but that our Gecko metric consistently performs best for Gecko(S)/(R) across tasks and on TIFA160. App. F.1 gives a thorough description of each task and intuitive examples for how they differ.

### 6.1 EXPERIMENTAL SETUP

**Metrics.** We benchmark two types of metrics. First, metrics that give a *single score*, including contrastive models (CLIP (Radford et al., 2021), PyramidCLIP (Gao et al., 2022) and 16 variants in Sec. F.4); (2) VNLI (Yarom et al., 2024); and (3) VQAScore (Lin et al., 2024). Second, interpretable QA/VQA based methods: TIFA (Hu et al., 2023), DSG (Cho et al., 2023a) and our metric Gecko.

**Back-end models.** For CLIP, we use a ViT-B/32 (Dosovitskiy et al., 2020) CLIP model and ViT-B/16 (Dosovitskiy et al., 2020) PyramidCLIP model. For VQAScore, we use a GeminiFlash (Reid et al., 2024) backend. For all the VQA-based metrics, we use PaLM-2 (Anil et al., 2023) as the LLM and PaLI (Chen et al., 2022) as the VQA models in all the metrics for fair comparison. When evaluating the Gecko metric, apart from using the LLM and VQA models above, we utilise a T5-11B model from Honovich et al. (2022) for NLI filtering and set the threshold $r$ at 0.005. This threshold was determined by examining QA pairs with NLI probability scores below 0.05. We observed that QAs with scores below 0.005 are typically hallucinations. We re-use the original prompts from TIFA for generating QAs, and add coverage notation to their selected texts as described in Sec. 5. We additionally explore how the performance of point-wise instance scoring changes for the Gecko metric if we swap out the QA/VQA models for a stronger Gemini Flash model. Finally, some baseline models are trained with a maximum text input length $L$,e.g. $L_{CLIP} = 77$ and $L_{VNLI} = 82$. For these models, we only take the first $L$ tokens from the text as input.

### 6.2 COMPARING AUTO-EVAL METRICS ON POINT-WISE INSTANCE SCORING

We first evaluate how well metrics measure T2I alignment at an instance level. We compute the Pearson and Spearman Ranked correlation between the auto-eval scores and human scores on all the instances in a prompt set. The evaluations are done on the Gecko benchmark and TIFA160.

| Metrics | QA | VQA | Spearman's $\rho$ | Kendall's $\tau$ |
|---------|-----|------|-------------------|------------------|
| ROUGE-L | | | 0.33 | 0.25 |
| METEOR | N/A | N/A | 0.34 | 0.27 |
| SPICE | | | 0.33 | 0.23 |
| CLIP | | | 0.33 | 0.23 |
| | GPT-3 | BLIP-2 | 0.56 | 0.44 |
| TIFA | GPT-3 | MPLUG | 0.60 | 0.47 |
| | PALM | PaLI | 0.43 | 0.32 |
| DSG | PALM | PaLI | 0.57 | 0.46 |
| Gecko | PALM | PaLI | **0.64** | **0.50** |

Table 6: **Comparing different metrics by their correlation with human Likert ratings on TIFA160.** The Gecko metric outperforms the others by a significant margin.

| Metrics | WL | Likert | SxS |
|---------|-----|--------|-----|
| | Pearson | | Acc |
| VideoCLIP | 0.18 | 0.21 | 28.0 |
| VQAScore$_{\text{Gemini Flash}}$ | 0.30 | 0.33 | 52.0 |
| Gecko$_{\text{Gemini Flash}}$ | **0.43** | **0.45** | **55.8** |

Table 7: **Correlation between auto-eval metrics and human ratings for text-to-video evaluations.** Gecko outperforms other auto-eval metrics on VBench overall consistency prompts, demonstrating the generality of the approach to other modalities.

**Component validation on proposed Gecko metric.** We validate the utility of the three key improvements we proposed: coverage, linear normalisation, and NLI filtering. Starting from our baseline TIFA, we include the improvements one at a time. The results in Table 5 uniformly demonstrate a positive impact. NLI filtering brings the largest boost among the three, underscoring the limitation of the PaLM-2 LLM in reliably generating high-quality and accurate QA pairs.

**Results on Gecko benchmark** We next compare auto-eval metrics. We start with metrics (CLIP and its variants, TIFA, DSG, Gecko) that do not rely on fine-tuning. As shown in Table 4, the Gecko metric outperforms other QA/VQA metrics using the same backend by a wide margin. Swapping out the backend of Gecko with a stronger GeminiFlash model leads to large improvements across the board. Contrastive models (e.g. CLIP variants) are worse than QA-based metrics, but VQAScore is a strong baseline. Finally, we compare Gecko with VNLI, our supervised baseline, as it is fine-tuned for text–image alignment on a mixed dataset containing COCO (which is used in Gecko(R)), while other metrics are zero-shot. It is worth noting that the correlation scores of different auto-eval metrics are generally higher on Gecko(S) than on Gecko(R). This validates that our skills-based benchmark has a more objective and balanced measure of alignment. We observe similar conclusions on the Gecko2K Reliable Prompts (Gecko2K-Rel) subset; results are in App. F.3.

**TIFA160 results.** We compare the Gecko metric with other metrics on TIFA160 (Hu et al., 2023), a set of 160 text–image pairs, each annotated with two Likert ratings. In Table 6, we list the results reported in Hu et al. (2023) and Cho et al. (2023a), and compare them with Gecko as well as our re-implementation of TIFA / DSG. Gecko has the highest correlation, with an average correlation 0.07 higher than that of DSG, when using the same QA and VQA models. This shows that the power of our proposed metric is from the method itself, not from the advance of models used.

**Skill-based evaluation with Gecko.** To better understand the differences between auto-eval metrics/annotation templates with respect to various skills, we visualise a breakdown of skills in Gecko(S) in Fig. 3 and App. E.1, F.5. The metrics have different strengths: e.g., we see that while Gecko, VQAScore, VNLI metrics are consistently good across skills, the Gecko metric is better on more complex and compositional language, DSG is best on compositional prompts, and VNLI is better on named entities. As with the overall results, these per skill conclusions seem to hold across templates.

**Qualitative examples.** We visualise examples in Fig. 4. For the negation example, the reason DSG(H) gives inconsistent results with WL/Likert here is that the question generation is confused by the negation (asking if there *are* cars as opposed to *no* cars). We can also see that VNLI and DSG mistakenly think none of the images are aligned. VNLI and DSG perform better on the shape prompt but VNLI scores Imagen incorrectly and DSG gives hard scores per question (0 or 1) and so it is sometimes not able to capture subtler differences in the human ratings.

### 6.3 COMPARING AUTO-EVAL METRICS ON PAIR-WISE INSTANCE SCORING

We measure how well an auto-eval metric is able to select between two generations given a prompt. We compare metrics' predictions with the human choices we collected by computing accuracy–the percentage of times the metric gets the comparison right. Results are in the *SxS* column in Table 4. Although Gecko was the clear winner on point-wise instance scoring, single-score metrics are generally very good at SxS comparison. PyramidCLIP was worse than TIFA and DSG on point-wise instance scoring, but it has a much higher SxS accuracy, showing that different human annotation templates *do not* always give the same result, and single-score metrics can be a good estimator on the pair-wise instance scoring task. While VQAScore is better than the Gecko metric on SxS com-

| Skill (subskill): Prompt: | lang/complexity (**negation**) A bridge with no cars on it. | | | | Shape: (**hierarchical**) The number 0 made of smaller circles | | | |
|---|---|---|---|---|---|---|---|---|
| | Imagen | Muse | SDXL | SD1.5 | Imagen | Muse | SDXL | SD1.5 |
| WL: | 1. | 1. | 1. | 1. | 1. | 1. | 0. | 0.67 |
| Likert: | 1. | 1. | 1. | 0.87 | 1. | 0.87 | 0.2 | 0.67 |
| DSG(H): | 0.5 | 0.5 | 0.5 | 0.67 | 0.92 | 1. | 0. | 0.89 |
| Gecko: | 0.96 | 0.94 | 0.93 | 0.91 | 0.9 | 0.95 | 0.55 | 0.75 |
| DSG: | 0.25 | 0.25 | 0.25 | 0.25 | 1. | 1. | 0. | 0. |
| VNLI: | 0.4 | 0.42 | 0.32 | 0.30 | 0.36 | 0.79 | 0.24 | 0.32 |

Figure 4: **Qualitative results.** Image generations of the four T2I models on prompts in Gecko(S), with the human annotation ratings and auto-eval scores.

parison on Gecko(S), Gecko is better on Gecko(R) and the Gecko metric is the only interpretable metric that has better or comparable performance with single-score metrics on SxS comparisons.

### 6.4 COMPARING AUTO-EVAL METRICS ON MODEL ORDERING

A good auto-eval metric should be able to give an overall model ordering for a set of prompts. To decide on a ground-truth ordering, we use Gecko2K-rel as it is the largest subset that has highest agreement across templates. We take the majority vote relationship in Fig. 13 as the ground truth. We compare these results to the significant relationships found using the auto-eval metrics in App. F.2 (we only use PaLM/PaLI-2 backends if there is a choice). We find that CLIP performs poorly, confusing wins with losses. All other auto-eval metrics perform well, never confusing a win with a loss but sometimes not finding significant relations when there is one or vice versa. Gecko correctly finds and predicts *all* significant relations, unlike the other metrics.

### 6.5 EXTENDING GECKO TO OTHER MODALITIES

To explore the generality of our approach on different modalities, we validate it on text-to-video generation. We choose a prompt set from VBench (Huang et al., 2024b) and compare the following models: Lumiere (Bar-Tal et al., 2024), Phenaki (Villegas et al., 2022) and WALT (Gupta et al., 2023). For human evaluation, we consider absolute (i.e., Likert, Word Level) and side-by-side templates. For automatic evaluation, we benchmark contrastive models (i.e., VideoCLIP; Xu et al. 2021) and VQA-based metrics. For VQA-based metrics, we extend the VQAScore and our fine-grained Gecko metric on videos using Gemini Flash, which can process long context multimodal inputs. We present the results in Table 7 and find that the Gecko metric agrees more closely with human judgement across all human templates than other metrics. See Appendix G for more details.

**Takeaway:** Although Gecko is the best metric on different human templates and modalities, we find that the ranking of different auto-eval metrics can change depending on whether they are evaluated on an instance-level template (e.g., Likert or DSG(H)), a comparative template (e.g. SxS) or for model ordering. It is important to evaluate metrics across a range of settings and in particular on one relative and one absolute template if under budget constraints.

## 7 CONCLUSIONS

We introduce the Gecko evaluation suite, a comprehensive set of prompts, human ratings across templates, and tasks to evaluate T2I models and alignment metrics. We find that looking at a single slice of the data (e.g., one annotation template, or one evaluation task) can give misleading observations of the relative benefits of one model or metric. Instead, we show that we need to use a comprehensive prompt set (or manually evaluated "reliable prompts") to achieve consistent model orderings and thereby confidence in model rankings. Given this evaluation suite, we demonstrate that our Gecko metric performs consistently best across three tasks, measuring how metrics perform in scoring each image–text instance with respect to their alignment as well as ranking models. Our work highlights the importance of standardising the evaluation framework with respect to the prompt sets, the annotation templates, and metrics used. This is crucial when conducting research on models and metrics, and also to make informed decisions.

## 8    ETHICS STATEMENT

When gathering our dataset, we ensure that raters are compensated and provide consent as described in App. D.1.1. We also run safety filters over the generated images before giving them to the raters. This work is a step towards better evaluation of text-to-image models which are known to halluci- nate. It gives tools to others developers and practitioners to properly understand and evaluate T2I models in the future.

## 9    REPRODUCIBILITY STATEMENT

We give extensive details of our setup in the Appendix. For human annotation, we visualise the templates used and give extensive detail on how these raw ratings are aggregated in App. D.1. For the dataset collation, we give the few shot prompts used to generate tags and templates: the few shot prompt for Gecko(R) is given in Listing 1. For Gecko(S), we give our full decomposition of skills in Table 8 with examples and an explanation of how we generated prompts for each specific skill in App. B.4 with sample few shot prompts. For metrics, we give full details of the baselines in Sec. 6.1 and the additional CLIP baselines in Sec. F.4. For Gecko metrics, we give the few shot prompt for generating coverage in Listing 3 and for generating the QAs in Listing 4.

**Acknowledgements**    We thank Zi Wang, Miloš Stanojević, and Jason Baldridge for their feedback throughout the project. We are grateful to Andrew Zisserman for his feedback on the manuscript. We thank Aayush Upadhyay and the rest of the Podium team for their help in running models.

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
