# APPENDICES

## A  OVERVIEW

In the Appendix, we give additional information on the benchmark, human annotation and corresponding results for T2I models, and experimental results for the auto-eval metrics.

**Gecko Benchmark:**    For the benchmark, we give further information on how we automatically tag Gecko(R) and semi-automatically generate prompts in Gecko(S) in App. B.1 and B.3 respectively. We then give more detail about the skill breakdown in Gecko(R) in App. B.2. We define and give examples for the sub-skills in Gecko(S) in App. B.4.

**Gecko Metric:**    We give further details on the Gecko metric in App. C.

**Human Annotation:**    For the human annotation, we give additional details of our setup including screenshots of the annotation templates used and qualitative limitations of each setup in App. D.1. We further discuss more experimental results comparing inter-annotator agreement and the raw predictions under each template in App. D.2. Finally, we visualise the most and least reliable prompts in App. D.3, giving an intuition for the properties of the prompt that lead to more or less agreement across templates.

**Additional results on T2I models:**    We give further results on using the annotated data to (1) compare T2I models by skill in App. E.1. We also compare how well prompts in TIFA160 are able to discriminate models under our human annotation setup in App. E.2 and find that they are less discriminative.

**Additional results for auto-eval metrics:**    We give an intuitive explanation of each task as well as how they can lead to different metric orderings in App. F.1. We then give additional results for the auto-eval metrics on Gecko2K and Gecko2K-rel, including more correlation results in App. F.3 but we find that conclusions are the same irrespective of how we compute correlation or using the reliable subset or full set. We give the raw results for the model-ordering evaluation in App. F.2 and results for different CLIP variants in App. F.4. Finally, we explore results per skill for different auto-eval metrics in App. F.5, give additional visualisations in App. F.6 and demonstrate that we can use Gecko to evaluate the per-word accuracy of the metric (this is not possible with other auto-eval metrics) in App. F.7.

## B  GECKO2K: MORE DETAILS

As described in Sec. 3.1, we use automatic tagging in order to tag prompts with different skills in Gecko(R). However, this has a few issues: (1) it can be error prone; (2) we are limited by the tagging mechanism in the skills that we tag; (3) we do not tag sub–skills. As a result, we devise a semi-automatic approach to build Gecko(S) by few-shot prompting an LLM, as discussed in Sec. 3.2 and curate a dataset with a number of skills and sub-skills for each skill. This dataset covers more skills and sub-skills than other datasets, as shown in Fig. 6, 7.

### B.1  AUTOMATIC TAGGING FOR GECKO(R)

As mentioned in Sec. 3.1, to obtain a better control for the skill coverage and prompt length, we resampled from the 10 datasets used in DSG1k (Cho et al., 2023a). To identify the categories covered in the prompts, we adopted an automatic tagging method similar to that used in DSG1K. This method utilizes a Language Model (LLM) to tag words in the text prompt, as shown in Listing 1. The only difference is that we also included named entities and landmarks to be the original categories, such as *whole*, *part*, *state*, *color* etc.

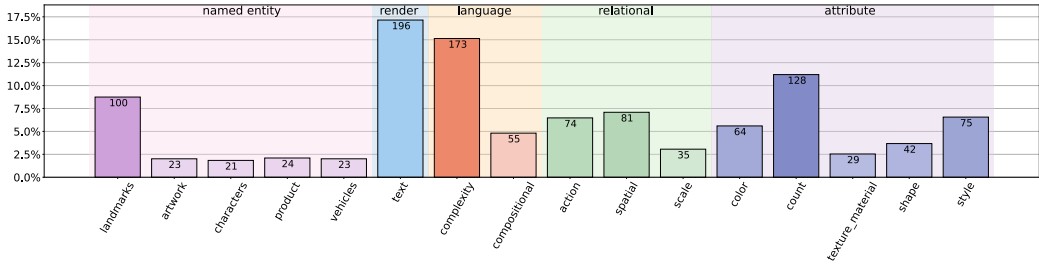

Figure 5: **Overview of Gecko(S).** The set of skills (coloured by the corresponding category) covered by the synthetic prompts. Note that we gather prompts by breaking each skill into sub-skills.

```
1
2  """
3  id: synthetic_v1_1
4  input: a man is holding an iPhone.
5  output: 1 | entity - whole (man)
6          2 | entity - named entity (iPhone)
7          3 | action - hold (man, iPhone)
8
9  id: diffusiondb_79
10 input: an hd painting by Vincent van Gogh. a bunch of zombified mallgoths hanging out at a hot topic store in
          the mall.
11 output: 1 | global - style (hd painting)
12          2 | global - style (Vincent van Gogh)
13          3 | entity - whole (mallgoths)
14          4 | attribute - state (mallgoths, zombified)
15          5 | other - count (mallgoths, ==bunch)
16          6 | entity - whole (hot topic store)
17          7 | entity - whole (mall)
18          8 | relation - spatial (mallgoths, hot topic store, at)
19          9 | relation - spatial (hot topic store, mall, in)
20 ...
21
22 id: {image_id}
23 input: {text_input}
24 output: {LLM_output}
25 """
```

Listing 1: The prompt used to automatically tag skills given text prompts from the base datasets in DSG1K in order to generate a more balanced Gecko(R).

## B.2 PROMPT DISTRIBUTION IN GECKO(R)

We resample 1000 prompts from the base datasets used in DSG1K and ensure a more uniform distribution over skills and prompt length. To sample more prompts featuring words from under-represented skills (e.g.TEXT RENDERING, SHAPE, NAMED IDENTITY and LANDMARKS), we use automatic tagging in App. B.1 to categorize the words in all prompts as pertaining to a given skill. We then resample, assigning higher weights to the under-represented skills. The resulting skill distribution is shown in Fig. 8. Although the resampling increases the proportion of under-represented skills, the overall distribution remains unbalanced. This underscores the necessity of acquiring a synthetic subset with a more controlled and balanced prompt distribution. To sample long prompts, we eliminate the constraint set in DSG1K (Cho et al., 2023a), which mandates that the sampled prompts should be shorter than 200 characters. This adjustment results in a more diverse prompt length distribution as shown in Fig. 8.

## B.3 TEMPLATES TO FEW-SHOT AN LLM FOR GECKO(S)

As discussed in Sec. 3.2, we semi-automatically create prompts for Gecko(S) by few-shot prompting an LLM. We give an example for the TEXT RENDERING skill in Listing 2. In short, we define a set of properties based on the sub-skills we want included in our dataset. In this case, we define *text length* and *language* (we use *English* and *Gibberish* but we note this could be easily extended to more languages). We then create examples that have those properties to create our few-shot prompt. We can query the LLM as many times as we like to create a distribution of prompts across different text lengths and languages. We do a similar setup for each of the skills and sub-skills we define below.

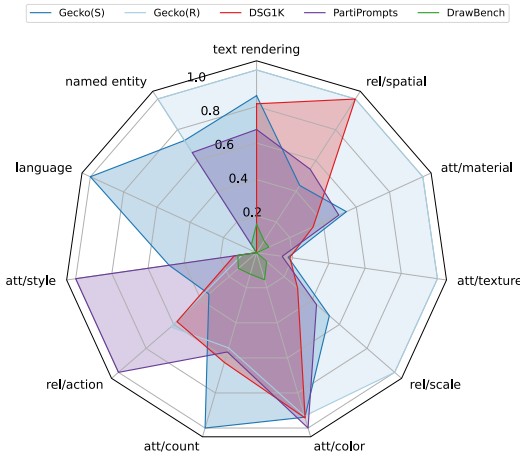

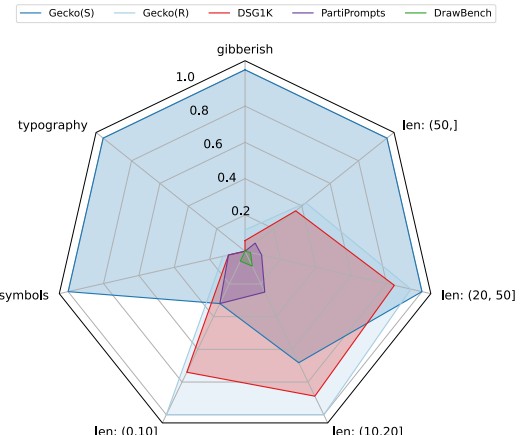

Figure 6: **Distribution of skills.** We visualise the distribution of prompts across different skills for Gecko(S)/(R), DSG1K (Cho et al., 2023a), PartiPrompts (Yu et al., 2022b) and DrawBench (Saharia et al., 2022). We use automatic tagging and, for each skill, normalise by the maximum number of prompts in that skill over all datasets. For most skills, Gecko2K has the most number of prompts within that skill.

Figure 7: TEXT RENDERING **skill.** We visualise the distribution of prompts across seven sub-skills explained in Table 2 ('len: ...' corresponds to bucketing different lengths of the text to be rendered). We normalise by the maximum number of prompts in the sub-skill (note that we only count unique texts to be rendered). The Gecko(S) dataset fills in much more of the distribution here than other datasets.

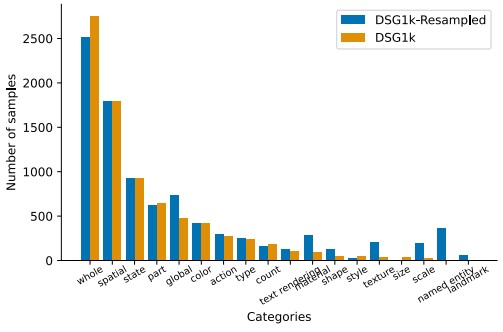

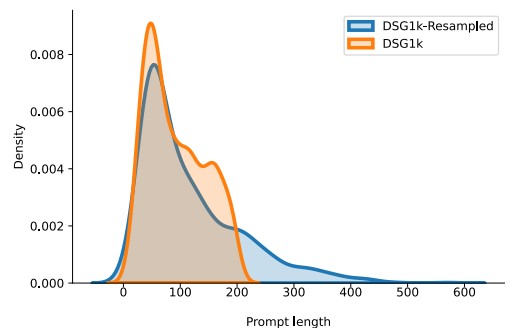

(a) The skill distribution (which is tagged at the word level).

(b) The prompt length distribution.

Figure 8: **Prompt distribution in DSG1K-Resampled(Gecko-R) and DSG1K.**

```
1  """
2  Generate captions for the given text of varying length. Be creative and imagine new settings.
3
4  Text length: 20
5  Language: English
6  Text: "look at that shadow!"
7  Caption: shadow of a stone, taken from the point of view of an ant, with the caption "look at that shadow!"
8
9  ...
10
11 Text length: {text_length}
12 Language: {language}
13 Text: {LLM_output}
14 Caption: {LLM_output}
15 """
```

Listing 2: Sample LLM template.

## B.4 BREAKDOWN BY SKILL/SUB-SKILL IN GECKO(S)

An overview of Gecko(S) is given in Fig. 5 and comparisons to other datasets for skills and a given subskill in Fig. 6,7. In this section we give more information on the skills and sub-skills within Gecko(S). We provide a detailed breakdown of each prompt sub-skill, including examples and justifications. Skills and sub-skills are listed in Table 8. We aim to cover semantic skills, some of which have already been covered in previous work (e.g.*shapes, colors* or *counts*), while further subdividing each skill to capture its different aspects and difficulty levels. By varying the difficulty of the prompts within a challenge we ensure we are testing the models and metrics at different difficulty levels and can find where models and metrics begin to break.

Each skill (such as SHAPE, COLOR, or NUMERICAL) is divided into sub-skills, so that prompts within that sub-skill can be distinguished based on difficulty or, if applicable, some other criteria that is unique to that sub-skill (i.e.prompts inspired by literature in psychology). We create a larger number of examples and subsample to create our final 1K set of prompts to be labelled.

### B.4.1 SPATIAL RELATIONSHIPS

This skill captures a variety of spatial relationships (such as *above, on, under, far from, etc.*) between two to three objects. In the most simple case, we measure a model's ability to understand common relationships between two objects. The difficulty is increased by combining simpler entities and requiring the ability to reason about implicit relationships. We use an LLM as described in Sec. B.3 to create these prompts, subsample and manually verify prompts are reasonable.

### B.4.2 ACTION

This skill examines whether the model can bind the right action to the right object, including unusual cases where we flip the subject and the object (i.e.*Reverse actions*). An example of a reverse setup is that we swap the entities in 'A penguin is diving while a dolphin swims' and create 'A penguin is swimming while a dolphin is diving'. We vary the difficulty by increasing the number of entities. We use an LLM as described in Sec. B.3 to create these prompts, subsample and manually verify prompts are reasonable.

### B.4.3 SCALE

We measure whether the model can reason about scale cues referring to commonly used descriptors such as *small, big* or *massive*. To reduce ambiguity, we typically refer to two objects, so that they can be compared in size. We test the ability to implicitly reason about scales by having *Comparative* prompts that contain several statements about objects, their relations and sizes. We use an LLM as described in Sec. B.3 to create these prompts, subsample and manually verify prompts are reasonable.

### B.4.4 COUNTING

The simplest sub-skill *Simple modifier* contains a number (digits such as "2", "3" or numerals such as "two", "three") and an entity. When selecting a vocabulary of words, we aimed to include words that occur less frequently in ordinary language (for example, "lemur" and "seahorse" occur less frequently than "dog" and "cat") (Speer, 2022). We focus on numbers 1—10, or 1—5 in more complex cases. Complexity is introduced by combining simple prompts containing just one attribute into compositional prompts containing several attributes. For example, simple prompts "1 cat" and "2 dogs" are combined into a single prompt "1 cat and 2 dogs" in the sub-skill *Additive*. We also test approximate understanding of quantities based on linguistic concepts of *many* and *few* in the *Quantifiers and negations* sub-skill.

### B.4.5 SHAPE

We test for basic and composed shapes, where composed shapes include objects arranged in a certain shape. *Hierarchial shapes* are of the following type: "The letter H made up of smaller letters S". This kind of challenge is used to study spatial cognition and the trade-off between global and local

| Skill | Sub-skill | Examples |
|---|---|---|
| Spatial Relationships rel/spatial | Simple | A cat above a dog.
The lemon is in the middle of the apples.
A bus is behind a truck going down the highway. |
| | Composed | The cat is near the banana. The banana is below the horse. The horse is on the truck. |
| Action rel/action | 1, 2, or 3 entities | A bear is running through a field.
A basketball is passed to a team member. |
| | Reverse actions | A ladybug is riding on the back of a flying unicorn.
A koala climbs a tree, an eagle stands on the branch and a penguin flies overhead. |
| Scale rel/scale | Single object | A small ship in a bottle.
A giant couch in a field. |
| | Comparative | A garlic is next to an onion and a tomato on a cutting board. The onion is larger than the garlic. The tomato is smaller than the garlic. |
| | Same size | The table is the same size as the cake.
The mouse is the same size as the dragon. |
| Counting att/count | Simple modifier | 2 cats.
Four lemurs. |
| | Additive | 5 burgers and one bonsai.
1 baobab, 2 cats and 3 dogs. |
| | Quantifiers and negations | Some shirts and some pizzas. There are more shirts than pizzas.
An image with fewer dogs than cats.
An image with no flowers in the vase. |
| Shape att/shape | Basic shapes | A line.
An octagon. |
| | Composed shapes | A star-shaped cookie.
Strawberries arranged in a heart shape. |
| | Hierarchical shapes | A square made of smaller letters g.
A smiley face made of strawberries. |
| Text Rendering render/text | Rendering | A creature with a clock shaped head with the words "flimflam, bishbash, gorp" written on it. The creature has a small body and two legs, and is pointing at the ground. There are two clocks on the ground, one showing the time of 7:24 and the other showing 3:30 p.m. |
| | Numerical | equation of "3+4 = 7" etched into a rock.
"Lorem ipsum dolor sit amet, $%&*(), 12345 + adipiscing elit" written on a chalkboard with a piece of chalk next to it. |
| | Font | "congratulations" written in fancy decorative cursive font on an antique 1920's typewriter.
"happiness" written in decorative font with a happy face next to it. |
| Color att/color | Simple colors | A pink salad.
A grey vase. |
| | Composed expressions | A pink unicorn, a white airplane, and a green potato.
A yellow couch and a green cookie. |
| | Colors and abstract shapes | A red rectangle on a green background.
A pink circle on a green background. |
| | Descriptive color terms | A pastel coloured train passing through the station.
A rainbow-colored bicycle leaning against a wall. |
| | Stroop | Text saying "yellow" in blue letters.
Text saying "black" in green letters. |

Table 8: Breakdown by skill and sub-skill including examples of prompts.

| Skill | Sub-skill | Examples |
|---|---|---|
| Surface Characteristics att/texture+material | Texture only | A fluffy floor in the bathroom.There is a silky fabric on a bumpy couch in the room. |
| | Material only | There is a metal lime in the bowl.A paper snake on the table. |
| | Combined | There is a soft floor made of wax and a shiny silver table in the room. A glossy diamond road. |
| Style att/style | style | A brightly colored canal in Venice, by Canaletto A cartoon of a cat by Goya. |
| | Visual medium | A sketch of a drawing of a flower in a pot. A glass vase in the style of el greco and the impressionists. |
| Language Complexity lang/complexity | Negation | A pencil sharpener without any pencils in it. A belt buckle with no belt. A leaflet with no text on it. |
| | Long prompt | An outdoors top-down view of purple sidewalk chalk on a concrete sidewalk reading, "Fear/ of/ Chores". The left side of the concrete slab has green algae growth that fades to the right. Small bits of smashed acorns are scattered across the slab. |
| | True paraphrase | The giraffe feeds the cat. The bystanders watch the dog. |
| | False paraphrase | The snake observes the kangaroo. The horse looks at the deer. |
| Compositional Language lang/compositional | Vary number of entities & attributes | An orange metal train. A plastic couch, a cyan blueberry, and 2 plates. 3 wooden pencils and 1 plastic fly. A brown plastic bus, a yellow plastic vase, and a yellow wooden salad. |
| Named Entities | Landmarks ne/landmarks | Ulvetanna Peak, Queen Maud Land, Antarctica during sunrise. Ashikaga Flower Park with beautiful wisteria in shades of violet and white. Burj Al Arab Jumeirah hotel with fireworks in the night and glowing water around. |
| | Animal Characters ne/characters | Grumpy Cat is sitting on a couch with a catnip toy. A cartoon of Laika playing in the snow. the lion cub named Simba is catching a ball. |
| | Vehicles ne/vehicles | A BMW M3 is on the road. A Ferrari is driving through roads in an Italian landscape. A Opel Ampera is upside down. |
| | Products ne/product | A bottle of Irn-Bru is sitting on a shelf. A Gucci bag with a red and white striped pattern. A Samsung Galaxy S III is on a car seat. |
| | Artwork ne/artwork | Charles IV of Spain and His Family is being painted on an easel. A painting of The Milkmaid hangs on the wall of a living room. A painting of Bacchus and Ariadne hanging in a stone building. |

attention in literature on higher-order cognition (Delis et al., 1986). We use an LLM as described in Sec. B.3 to create these prompts, subsample and manually verify prompts are reasonable.

### B.4.6 TEXT RENDERING

We investigate a model's ability to generate text (both semantically meaningful and meaningless), including text of different lengths. We further test for the ability to generate symbols and numbers, as well as different types of fonts. We use an LLM as described in Sec. B.3 to create these prompts, subsample and manually verify prompts are reasonable.

### B.4.7 COLOUR

The simplest prompts in this skill include basic colours bound to objects. As before, we introduce complexity by combining several simpler prompts (either two or three objects bound with a colour attribute). To include diversity of possible colour attributes, we also test descriptive colour terms such as "pastel" or "rainbow-coloured". Finally, the sub-skill *stroop* contains prompts of the type "Text saying 'blue' in green letters" similar to the incongruent condition in the Stroop task (Stroop, 1935) used to study interference between different cognitive processes.

### B.4.8 SURFACE CHARACTERISTICS

Surface characteristics include texture and material. We first test for each sub-skill individually, and then combined. Generally, some prompts in this skill can be difficult to visualise as they might include descriptions that are typically of tactile nature ("abrasive" or "soft").

### B.4.9 STYLE

We divide prompts into two sub-skills: one depicting a style of an artist, and another to capture different visual mediums (such as *photo, stained glass* or *ceramics*). We use an LLM as described in Sec. B.3 to create these prompts, subsample and manually verify prompts are reasonable.

### B.4.10 NAMED ENTITY

This skill evaluates a model's knowledge of the world through named entities, focusing on specific entity types such as *landmarks*, *artwork*, *animal characters*, *products*, and *vehicles*, which are free of personally identifiable information (PII).

For the landmark class, we choose landmarks from the Google Landmarks V2 dataset and ensure we cover different continents and choose landmarks with high popularity (Weyand et al., 2020). Given this set of landmarks, we use an LLM as described in Sec. B.3 to create these prompts, subsample and manually verify prompts are reasonable.

For the other classes, to curate diverse named entities, we first gather candidates from Wikidata using SPARQL queries. A simple query (e.g., `instance of (P31)` is `painting (Q3305213)`) might yield an excessively large number of candidates. Therefore, we impose conditions to narrow down the query responses. See our criteria below.

*Artwork* : Created before the 20th century; any media; any movement

*Animal Characters* : Anthropomorphic/fictional animals; real animals with names

*Products* : Electric devices; food/beverage; beauty/health

*Vehicle* : Automobiles; aircraft

Once we have a candidate set for each entity class, we focus on selecting reasonably popular entities that are widely recognised and appropriate to present to models. We assess popularity using the number of incoming links to and contributors on their English Wikipedia pages as proxies (Geva et al., 2021). Finally, we manually curate the final set of named entities, selecting them based on their ranked popularity scores.

### B.4.11 LANGUAGE COMPLEXITY

We evaluate models on prompts with "tricky" language structure / wording. For this skill, we include 4 sub-skills: *negation*, *long prompt*, and *true / false phrases*. We sampled 19 prompts for *negation* from LVIS (Gupta et al., 2019), COCO stuff (Caesar et al., 2018), and MIT Places (Zhou et al., 2017); 58 *true paraphrases* and 58 *false paraphrases* from BLA (Chen et al., 2023); and finally crowdsourced 38 for *long prompt* with the help from English major raters. It should be noted that while we do not cover the entire spectrum of complex language (e.g. passives, coordination, complex relative clausals, etc.), the subcategories included cover the most prominent pain points of image generation models per our experimentation.

We also include a language complexity metric which can be run over all prompts. Here we treat language complexity from two perspectives – semantic and syntactic.

- *Semantic complexity*. The quantity of semantic elements included in a prompt.
- *Syntactic complexity*. The level of complexity of the syntactic structure of a prompt.

Concretely, we define *semantic complexity* as the number of entities extracted from a prompt. Taking the visual relevance of the task into account, we apply *Stanford Scene Graph Parser* (Schuster et al., 2015) for entity extraction and count the number of unique entities as the proxy for semantic complexity. For *syntactic complexity*, we implement a modified variant of Ohta et al. (2013) to look for the deepest central branch in the dependency tree of a prompt (pseudo-code below) [§] to gauge the complexity of its syntactic structure.

```
1  def central_depth(node) -> Tuple[int, int]:
2    return (max(central_depth(child)[1]+1 in node.children if child.position < node.position),
3            max(central_depth(child)[0]+1 in node.children if child.position > node.position))
```

## C GECKO METRIC: MORE DETAILS

### C.1 LLM PROMPTING FOR GENERATING COVERAGE

```
1   """
2   Given a image description, label the visually groundable words in the description, and a score indicating how
        visually groundable it is.
3   Classify each word into a type (entity, activity, attribute, counting, color, material, spatial, location,
        shape, style, other).
4
5   Description:
6   Portrait of a gecko wearing a train conductor's hat and holding a flag that has a yin-yang symbol on it.
        Woodcut.
7   The visual-groundable words and their scores are labelled below:
8   {1}[Portrait, style, 0.8] of {2}[a, count, 1.0] {3}[gecko, entity, 1.0] {4}[wearing, activity, 1.0] {5}[a,
        count, 1.0] {6}[train conductor's hat, entity, 1.0] and {7}[holding, entity, 1.0] {8}[a, count, 1.0]
        {9}[flag, entity, 1.0] that has {10}[a yin-yang symbol, entity, 1.0] on it. {11}[Woodcut, material,
        1.0].
9
10
11  Description:
12  square blue apples on a tree with circular yellow leaves
13  The visual-groundable words and their scores are labelled below:
14  {1}[square, shape,1.0] {2}[blue, color, 1.0] {3}[apples, entities, 1.0] {4}[on, spatial, 1.0] {5}[a, count,
        1.0] {6}[tree, entity, 1.0] with {7}[circular, shape, 1.0] {8}[yellow, color, 1.0] {9}[leaves, entity,
        1.0]
15
16  Description:
17  A small dog running on a beach happily on a sunny day
18  The visual-groundable words and their scores are labelled below:
19  {1}[A, count, 1.0] {2}[small, attribute, 0.3] {3}[dog, entity, 1.0] {4}[running, activity, 1.0] {5}[on,
        spatial, 1.0] {6}[a, count, 0.0] {7}[beach, entity, 1.0] happily on {8}[a, count, 0.0] {9}[sunny,
        attribute, 0.5] day.
20
21  Description:
22  acrylic drawing, illustration, multiple mushrooms and pink jello, naive, flat, sketchy, purple background.
23  The visual-groundable words and their scores are labelled below:
24  {1}[crylic drawing, style, 1.0], {2}[illustration, style, 0.2], {3}[multiple, count, 0.5] {4}[mushrooms,
        eneity, 1.0] and {5}[pink, color, 1.0] {6}[jell, entity, 1.0], {7}[naive, attribute, 0.1], {8}[flat,
        attribute, 0.8] {9}[sketchy, style, 0.8] {10}[purple, color, 1.0] {11}[background, entity, 1.0].
25
26  Description:
27  a girl with many braids, riding away on her bike through the city, children's book cover illustration,
        detailed background, vibrant colors
```

---

[§]As an implementation note, we implemented Schuster et al. (2015) and Ohta et al. (2013) with SpaCy 2 (Honnibal & Montani, 2017) as the workhorse parsing backend.

```
28 The visual-groundable words and their scores are labelled below:
29 {1}[a, count, 1.0] {2}[girl, entity, 1.0], with {3}[many, counts, 0.8] {4}[braids, entity, 1.0], {5}[riding
        away, activity, 1.0] on her {6}[bike, entity, 1.0] through the {7}[city, place, 0.8], {8}[children's
        book cover illustration, style, 0.8], {9}}[detailed background, entity, 1.0] {10}[vibrant colors, colors
        , 1.0].
30
31 Description:
32 """
```

Listing 3: Sample LLM template for generating word coverage.

## C.2 LLM PROMPTING FOR GENERATING QAs

```
1  """
2  Given a image description, generate one or two multiple-choice questions that verifies if the image
        description is correct.
3  Classify each concept into a type (object, human, animal, food, activity, attribute, counting, color, material
        , spatial, location, shape, other), and then generate a question for each type.
4
5  Description:
6  A man posing for a selfie in a jacket and bow tie.
7  The visual-groundable words and their scores are labelled below:
8  A {1}[Man, human] {2}[posing, activity] for a {3}[selfie, object] in a {4}[jacket, object] and a {5}[bow tie,
        object].
9  Generated questions and answers are below:
10 About {1}:
11 Q: is there a man in the image?
12 Choices: yes, no
13 A: yes
14 About {2}:
15 Q: is the man posing for the selfie?
16 Choices: yes, no
17 A: yes
18 About {3}:
19 Q: is the man taking a selfie?
20 Choices: yes, no
21 A: yes
22 About {4}:
23 Q: is the man wearing a jacket?
24 Choices: yes, no
25 A: yes
26 About {5}:
27 Q: is the man wearing a bow tie?
28 Choices: yes, no
29 A: yes
30
31
32 Description:
33 A horse and several cows feed on hay.
34 The visual-groundable words and their scores are labelled below:
35 A {1}[horse, animal] and {2}[several, count] {3}[cows, animal] {4}[feed, activity] on a {5}[hay, object].
36 Generated questions and answers are below:
37 About {1}:
38 Q: is there a horse?
39 Choices: yes, no
40 A: yes
41 About {2}:
42 Q: are there several cows?
43 Choices: yes, no
44 A: yes
45 About {3}:
46 Q: are there cows?
47 Choices: yes, no
48 A: yes
49 About {4}:
50 Q: are the horse and cows feeding on hay?
51 Choices: yes, no
52 A: yes
53 About {5}:
54 Q: is there hay?
55 Choices: yes, no
56 A: yes
57
58 Description:
59 ...
60
61 Description:
62 """
```

Listing 4: Sample LLM template for generating QAs.

## C.3 DISCUSSION

Here we discuss the potential limitations of the models we rely on and how we mitigate those issues, as well as give quantitative results around how impactful those potential issues are in practice.

Note that we treat these models as black boxes (we do not consider fine-tuning or further calibration) which gives the benefit, as shown in Table 4, that as models improve (e.g. by swapping `PALM-2/PALI` for `Gemini Flash`), so too will our metric, with no further effort.

**Potential issue 1: Hallucination of the QA model.** The QA model could hallucinate, generating erroneous questions that are not grounded in the prompt, leading to worse performance. There are two factors that help mitigate this: (1) the use of the NLI model and (2) the ability of the LLM used to not hallucinate in the first place. We quantify the accuracy of the NLI model. To do so, we randomly chose 1.8K question/answer pairs from Gecko(R)/(S) and annotated whether they are hallucinations or not. We find that the NLI model is ∼93% accurate on the `PALM-2` setup. The utility of the NLI model is further validated in Table 5, which shows that adding NLI filtering improves results. We also evaluate how often the NLI model removes questions for the older `PALM-2` model as opposed to `Gemini Flash`. We find that the NLI model removes 13% of questions for `PALM-2` but only 2% for `Gemini Flash`, indicating that a better model will hallucinate less. This result validates the finding in Table 4 that as models improve, so too does our metric. In all, we find that hallucination can be mitigated effectively through the use of the NLI model and that with better LLMs, the impact of this issue will be diminished.

**Potential issue 2: Bias of the VQA model.** The VQA model could be biased or give poor scores. We note several factors that indicate the scores are useful and that these models do not suffer from severe bias. First, Cho et al. (2023a) have validated that `PALI` and even weaker models achieve high accuracy on such VQA style questions. We also use the largest model – `PaLI-17B`; prior work has found that versions with the largest language component are better calibrated (Kostumov et al., 2024). Second, we break down the VQA score into multiple questions and so we are more robust to incorrect scores arising from a single question. Third, we check for a strong 'yes' bias, which has been found in prior work evaluating VQA models (Agrawal et al., 2018) and is relevant, as many of our QAs are yes/no questions due to the few-shot prompt. We evaluate how `Gemini Flash` responds to 'blind' questions: given no image and a question, will the VQA model always output a given answer. We find that we obtain 20% 'yes' answers and 80% 'no', indicating that the model does not have a strong 'yes' bias. We also note that if a model is biased, it is equally biased for any input, which means that the relative comparison is valid.

Finally, our comprehensive results demonstrate that this potential bias is *not* a problem. First, we ablate the utility of the scoring component in Table 5 and find that it improves results. Second, we find that our metric performs well, obtaining 72/79% agreement with human preference and on average 0.53 correlation (see Table 4). If the VQA model were terribly biased, it would *ignore* visual input, leading to chance performance on the pair-wise instance scoring task. Thus the high performance on our comprehensive benchmark, as well as our ablations, demonstrate the utility of leveraging the scores from the VQA models.

# D   HUMAN ANNOTATION: MORE DETAILS AND EXPERIMENTS

## D.1   ANNOTATION TEMPLATES

**Likert scale.** We follow the template of Cho et al. (2023a) and collect human judgements using a 5-point Likert scale by asking the annotators "How consistent is the image with the prompt?" where *consistency* is defined as how well the image matches the text description. Annotators are asked to choose a rating from the given scale, where 1 represents *inconsistent* and 5 *consistent*, or a sixth *Unsure* option for cases where the text prompt is not clear. Choosing this template enables us to compare our results with previous work, but does not provide fine-grained, word-level alignment information. Moreover, while Likert provides a simple and fast way to collect data, challenges such as defining each rating especially when used without textual description (e.g., what 2 refers to in terms of image–text consistency), can lead to subjective and biased scores (Heo et al., 2022; Liang et al., 2020).

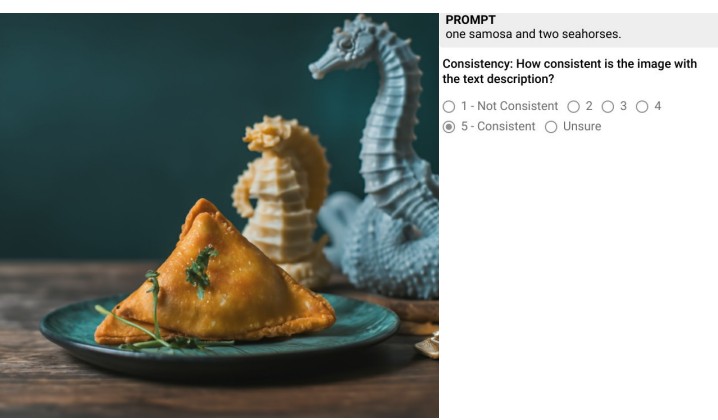

Figure 9: **Likert annotation template user interface.** Example depicting the interface shown to the annotators when performing evaluation tasks with the Likert template. Raters are given the prompt and image and asked to rate on a 5-point scale how consistent the image is with respect to the prompt. An *Unsure* option is also given the annotators.

**Word-level alignment (WL).** To collect word-level alignment annotations, we use the template of Liang et al. (2023) and define an overall image–text alignment score using the word-level information. Given a text–image pair, raters are asked to annotate each word in the prompt as *Aligned*, *Unsure*, or *Not aligned*. Note that for each text–image pair under the evaluation, the number of effective annotations a rater must perform is equal to the number of words in the text prompt. Although potentially more time consuming than the Likert template, we find that annotators spend ∼30s more to rate a prompt–image pair with WL than Likert.

We compute a score for each prompt–image pair per rater by aggregating the annotations given to each word. A final score is then obtained by averaging the scores of 3 raters.

**DSG(H).** We also use the annotation template of Cho et al. (2023a) that asks the raters to answer a series of questions for a given image, where the questions are generated automatically for the given text prompt as discussed in Cho et al. (2023a).

In addition, raters can mark a question as *Invalid* in case a question contradicts another one. The total number of *Invalid* ratings per evaluated generative model is given in App. D.1. Annotators could also rate a question as *Unsure*, in cases where they do not know the answer or find the question subjective or not answerable based on the given information.

For a given prompt, the number of annotations a rater must complete is given by the number of questions. We calculate an overall score for an image–prompt pair by aggregating the answers across all questions, and then averaging this number across raters to obtain the final score.

**Side-by-side (SxS).** We consider a template in which pairs of images are directly compared. The annotators see two images from two models side-by-side and are asked to choose the image that

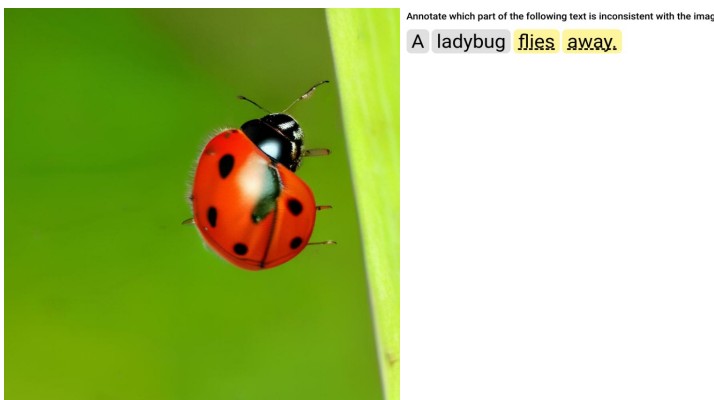

Figure 10: **Word-level annotation template user interface.** Example depicting the interface shown to the annotators when performing evaluation tasks with the WL template. Raters are asked to click on words they find are not aligned with image, and double click on the words where they are unsure.

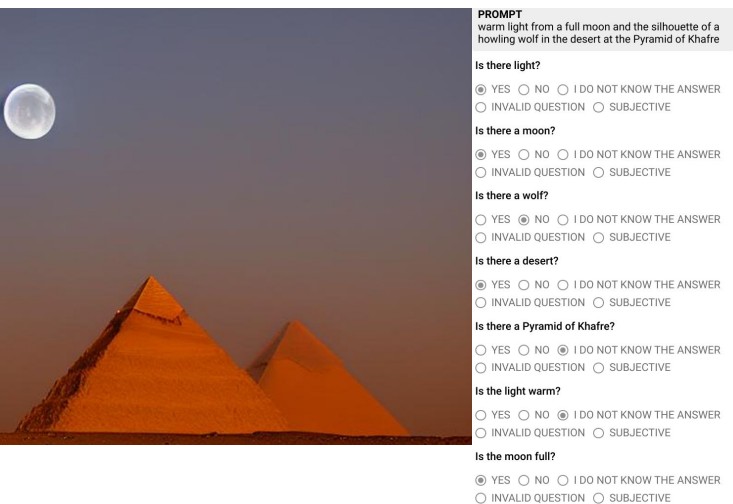

Figure 11: **DSG(H) annotation template user interface.** Example depicting the interface shown to annotators when performing evaluation tasks with the DSG(H) template. Raters are given the image, prompt, and respective automatically generated questions. There are 5 options for answering each question. In our analysis, both *I do not know the answer* and *Subjective* answers are considered as *Unsure*.

*is better aligned* with the prompt or select *Unsure*. We obtain a score for each comparison by computing the majority voting across all 3 ratings. In case there is a tie, we assign *Unsure* to the final score of an image–prompt pair.

### D.1.1 DATA COLLECTION DETAILS.

We recruited participants (N = 40) through a crowd-sourcing pool. The full details of our study design, including compensation rates, were reviewed by our institution's independent ethical review committee. All participants provided informed consent prior to completing tasks and were reimbursed for their time. Considering all four templates, both Gecko subsets, and the four evaluated generative models, approximately 108K answers were collected, totalling 2675 hours of evaluation.

### D.1.2 PERCENTAGE OF UNSURE RATINGS FOR EACH ANNOTATION TEMPLATE/MODEL

One of the innovations of our human evaluation setup is to allow for annotators to reflect uncertainty in their ratings. In Table 9 we show the percentage of *Unsure* ratings for each absolute comparison

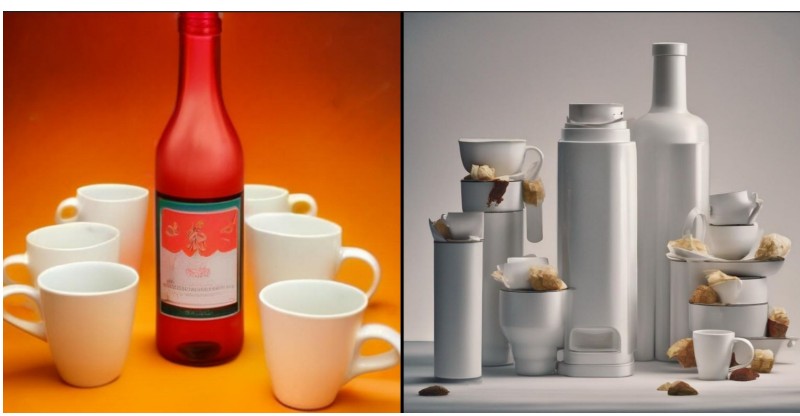

Figure 12: **Side-by-side comparison annotation template user interface.** Example depicting the interface shown to the annotators when performing evaluation tasks with the side-by-side template. Raters are given a pair of images from different models, the prompt used to generate them and asked to pick which one is more consistent with the prompt. An *Unsure* option is also given.

Figure 13: **Comparing models using human annotations.** We compare model rankings on Gecko(S), Gecko(R), their reliable subsets G(S)-rel, G(R)-rel and both subsets (G2K-rel). Each grid represents a comparison between two models. Entries in the grid depict results for WL, Likert (L), DSG(H) (D(H)), and side-by-side (SxS) scores. The > sign indicates the left-side model is better, worse (<), or not significantly different (=) than the model on the top. Green indicates cases where all results were the same across templates, yellow where templates didn't disagree with each other, and red cases where at least template disagreed with others.

annotation template. Overall, we find that evaluations with Gecko(R) yield a higher percentage of *Unsure* ratings in comparison to Gecko(S).

| Gen. model | WL | | Likert | | DSG(H) | |
|---|---|---|---|---|---|---|
| | Gecko(R) | Gecko(S) | Gecko(R) | Gecko(S) | Gecko(R) | Gecko(S) |
| Imagen | 43.52 | 18.46 | 20.25 | 2.07 | 30.90 | 29.55 |
| Muse | 41.09 | 23.91 | 18.10 | 2.42 | 33.47 | 32.65 |
| SDXL | 20.09 | 20.94 | 4.24 | 2.05 | 31.77 | 29.64 |
| SD1.5 | 13.35 | 26.48 | 10.04 | 4.09 | 37.02 | 33.08 |

Table 9: **Percentage of *Unsure* ratings.** Overall, evaluation with Gecko(S) yields fewer *Unsure* ratings across all models and templates.

## D.2 ADDITIONAL EXPERIMENTAL RESULTS

### D.2.1 PAIRWISE MODEL COMPARISONS WITH RELIABLE PROMPTS

### D.2.2 CORRELATION ACROSS TEMPLATES AND MODELS.

We show the correlation between templates and models for both Gecko(R) and Gecko(S) in Table 10.

| Gen. models | Gecko(R) | | | | | | Gecko(S) | | | | | |
|---|---|---|---|---|---|---|---|---|---|---|---|---|
| | Likert vs WL | | Likert vs DSG(H) | | WL vs DSG(H) | | Likert vs DSG(H) | | Likert vs WL | | WL vs DSG(H) | |
| | Pearson | Spearman | Pearson | Spearman | Pearson | Spearman | Pearson | Spearman | Pearson | Spearman | Pearson | Spearman |
| SD1.5 | 0.56 | 0.64 | 0.56 | 0.60 | 0.57 | 0.65 | 0.60 | 0.61 | 0.60 | 0.62 | 0.74 | 0.76 |
| SDXL | 0.60 | 0.63 | 0.52 | 0.57 | 0.56 | 0.61 | 0.60 | 0.50 | 0.56 | 0.62 | 0.78 | 0.79 |
| Muse | 0.67 | 0.62 | 0.61 | 0.63 | 0.61 | 0.66 | 0.51 | 0.52 | 0.51 | 0.53 | 0.77 | 0.75 |
| Imagen | 0.65 | 0.67 | 0.59 | 0.62 | 0.68 | 0.71 | 0.63 | 0.57 | 0.59 | 0.62 | 0.81 | 0.80 |

Table 10: **Correlation between all absolute comparison templates.** We compute Pearson and Spearman correlation coefficients for all pairs of templates for both Gecko(R) and Gecko(S). We find significant results with $p < 0.001$ for all cases and that scores of all metrics are at least moderately correlated, with the finer-grained templates, WL and DSG(H), being more correlated with each other in comparison to Likert.

### D.2.3 DISTRIBUTION OF SCORES PER PROMPT-IMAGE PAIRS ACROSS ANNOTATION TEMPLATES.

We plot the distribution of scores per each evaluated prompt-image pair for all the absolute comparison templates. The violin plots in Fig. 14-15 show the distributions for Gecko(R) and Gecko(S), respectively. It is possible to notice that scores obtained for Muse with WL and DSG(H) are more concentrated in values closer to 1 for both templates, corroborating findings from Sec. 4.2 where results showed Muse was the overall best model.

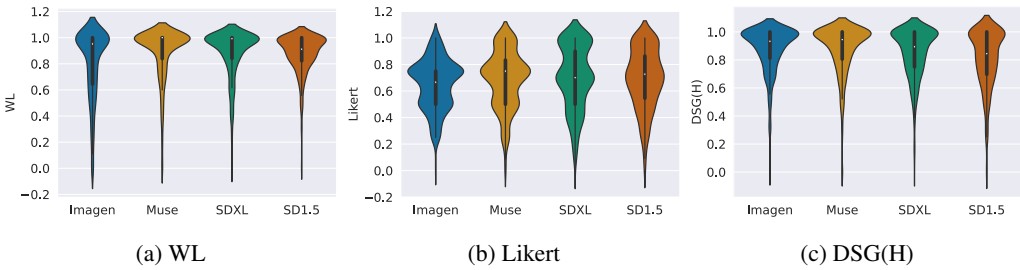

|     (a) WL     |     (b) Likert     |     (c) DSG(H)     |

Figure 14: **Distribution of scores for Gecko(R).** We show violin plots for scores obtained with all absolute comparison templates.

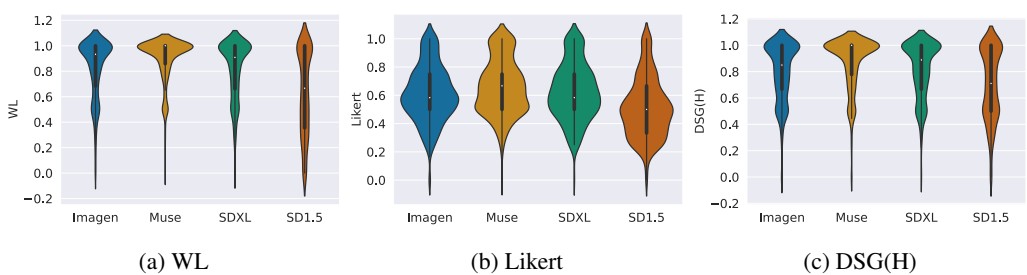

|     (a) WL     |     (b) Likert     |     (c) DSG(H)     |

Figure 15: **Distribution of scores for Gecko(S).** We show violin plots for scores obtained with all absolute comparison templates.

### D.2.4 SIDE-BY-SIDE TEMPLATE.

In Table 11 we show inter-annotator agreement results for the side-by-side annotation template for Gecko(R) and Gecko(S), along with the respective difference in agreement when using only the reliable prompts for both Gecko2K subsets. In Table 12 we present the results of the comparison between the side-by-side template and the absolute comparison ones.

|  | Gecko(R) | Gecko(R)-rel | $\Delta$ | Gecko(S) | Gecko(S)-rel | $\Delta$ |
|---|---|---|---|---|---|---|
| Imagen vs Muse | 0.438 | 0.465 | 0.027 | 0.485 | 0.625 | 0.140 |
| Imagen vs SDXL | 0.440 | 0.425 | -0.015 | 0.521 | 0.608 | 0.087 |
| Imagen vs SD1.5 | 0.402 | 0.431 | 0.029 | 0.581 | 0.652 | 0.071 |
| Muse vs SDXL | 0.471 | 0.489 | 0.018 | 0.570 | 0.638 | 0.068 |
| Muse vs SD1.5 | 0.539 | 0.592 | 0.053 | 0.600 | 0.617 | 0.017 |
| SDXL vs SD1.5 | 0.389 | 0.438 | 0.049 | 0.522 | 0.562 | 0.040 |

Table 11: **Side-by-side template: inter-annotator agreement.** We compute Krippendorff's $\alpha$ for Gecko(R) and Gecko(S) and the difference ($\Delta$) in $\alpha$ when using only reliable prompts for both subsets of Gecko2K. In both cases, using the reliable subsets increases the overall inter-annotator agreement.

|  | WL | Likert | DSG(H) |
|---|---|---|---|
| Imagen vs Muse | 0.736 | **0.755** | 0.728 |
| Imagen vs SDXL | 0.690 | 0.705 | **0.732** |
| Imagen vs SD1.5 | 0.672 | 0.632 | **0.708** |
| Muse vs SDXL | 0.704 | **0.746** | 0.703 |
| Muse vs SD1.5 | 0.759 | 0.740 | **0.749** |
| SDXL vs SD1.5 | 0.693 | **0.700** | 0.689 |
| Average | 0.709 | 0.713 | **0.718** |

Table 12: **Comparing side-by-side and absolute templates on Gecko2K-rel.** We compare the side-by-side template with the absolute comparison ones by computing the accuracy obtained by WL, Likert, and DSG(H) scores when using them to compare pairs of images on Gecko2K-rel. In this case, the ground-truth is assumed to be the results obtained with the side-by-side template.

### D.3 RELIABLE PROMPTS: EXAMPLES OF IMAGE-PROMPT PAIRS WITH HIGH HUMAN (DIS)AGREEMENT

In this section we show a representative list of prompts and corresponding images where human annotators were most likely to either agree or disagree in their ratings. The annotators agreed in ratings if they gave similar scores across for an image-text pair, meaning that the resulting mean variance was zero or close to zero. We refer to such prompts as "high agreement" prompts. In contrast, if annotators gave different ratings for a text-image pair, this would result in higher mean variance and we call such prompts "high disagreement" prompts.

To find prompts with high agreement across raters for all templates and all models, for each model-template combination we pick a subset of responses with low variance. Low variance is defined as the mean variance of a prompt-image pair for a model-template pair being below a certain threshold. The threshold is set as 10% of the maximum variance for that model-template set of ratings for both Gecko(R) and Gecko(S). Analogously, we also find a set of prompts with high disagreement; for this we find prompts that have mean variance above 1% of the maximum variance for a given template and for prompts from Gecko(R) and Gecko(S). The specific threshold value here is relevant only insofar as it captures at least 10 prompt-image pairs which we are interested in visualising. Then, we find prompts with high agreement by intersecting all model-template prompt sets where prompts have been selected based on the threshold. The procedure is analogous for low agreement prompts. For Gecko(R), both sets, namely the set of prompts with high agreement as well as the set of prompts with high disagreement have 34 prompts each. For Gecko(S), the set of prompts with high agreement has 62 prompts, while the set of prompts with high disagreement contains 85 prompts. A subset of 10 prompts for all different combinations is listed in Tables 13-16 and corresponding images are shown in the Figure 16-19.

Based on the analyses of such subsets, we observe several interesting trends. First, for Gecko(R) the prompts with higher agreement tend to be significantly shorter in length ($\mu = 54.32, \sigma = 32.18$) as measured by the number of characters, compared to the length of prompts with high disagreement ($\mu = 173.35, \sigma = 86.35$, Welch's t-test $t(41.99) = -7.42$ (p<0.001). The same observation holds for Gecko(S), where high agreement prompts were also significantly shorter ($\mu = 20.77, \sigma = 18.03$), than high disagreement prompts ($\mu = 82.48, \sigma = 95.17$, Welch's t-

| | **High Agreement Prompts (Gecko(R))** |
|---|---|
| 1 | three men riding horses through a grassy field |
| 2 | a small bathroom with a shower and a toilet |
| 3 | a wooden table with four wooden chairs in front of two windows |
| 4 | a large colgate clock is by the water |
| 5 | a slice of chocolate cake is on a small plate |
| 6 | a black cat sitting in a field of grass |
| 7 | The Statue of Liberty made of gold |
| 8 | a man riding skis down a snow covered slope |
| 9 | a vast, grassy field with animals in the distance |
| 10 | a plastic bento box filled with rice, vegetables and fresh fruit |

Table 13: Selected prompts with a high level of agreement in scores among raters for Gecko(R).

test $t(92.20) = -5.80$ (p<0.001). We further observe that prompts where raters tend to agree more are highly specific (i.e.they refer to one or just a few objects with few attributes), whereas prompts with high disagreement tend to describe more complex scenes with visual descriptors and often mentioning named entities or text rendering. Intuitively, this makes sense as longer prompts are more likely to require several skills.

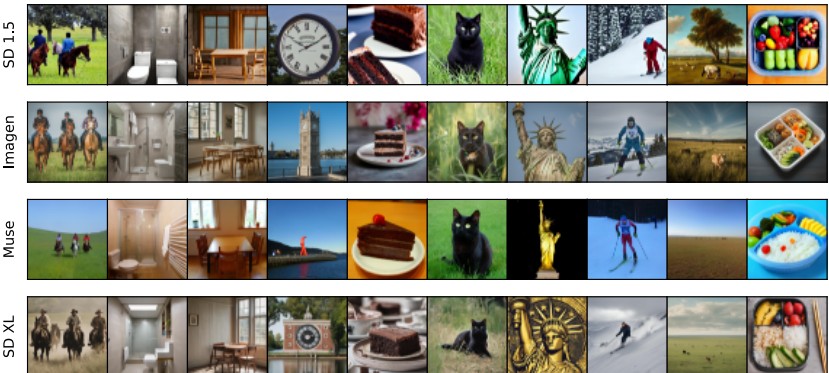

Figure 16: Images generated from Gecko(R) prompts with high level of agreement among raters. Prompts are listed in Table 13.

| | **High Agreement Prompts (Gecko(S))** |
|---|---|
| 1 | A star-shaped cookie |
| 2 | a pastel coloured train passing through the station. |
| 3 | a green boat. |
| 4 | the cat wears a gray shirt and holds a frisbee |
| 5 | a red motorcycle. |
| 6 | a black fish. |
| 7 | a pink bottle. |
| 8 | two mushrooms. |
| 9 | five cats. |
| 10 | a dog named Balto is running on a beach. |

Table 14: Selected prompts with a high level of agreement in scores among raters for Gecko(S).

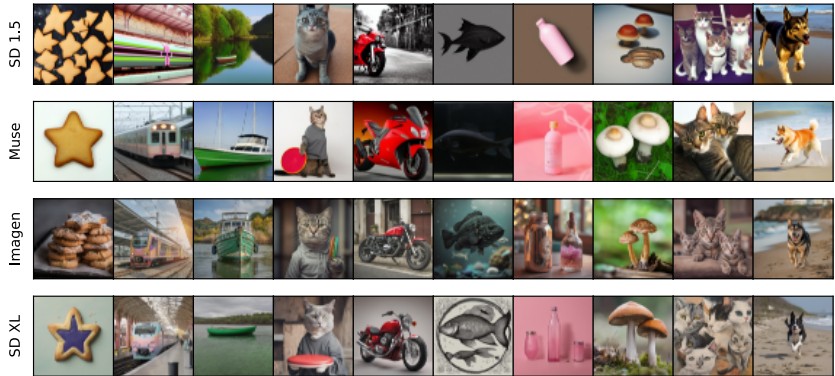

Figure 17: Images generated from Gecko(S) prompts with high level of agreement among raters. Prompts are listed in Table 14.

| High Disagreement Prompts (Gecko(R)) |
| --- |
| 1    Studio shot of sculpture of text 'cheese' made from cheese, with cheese frame. |
| 2    vintage light monochrome six round and oval label set Illustration |
| 3    There is a person snow boarding down a hill. There are tracks in the snow all around the snowboarder. There is a large rock in the snow next to them. There is a green pine tree in front of the snowboarder. The snowboarder is wearing blue ski pants and a blue and yellow jacket. They have a yellow snowboard on their feet. |
| 4    pillow in the shape of words 'ready for the weekend', letterism, funny jumbled letters, [ closeup ]!!, breads, author unknown, flat art, swedish, diaper-shaped, 2000, white clay, surreal object photography |
| 5    a sunflower field with a tractor about to run over a sunflower, with the caption 'after the sunflowers they will come for you' |
| 6    a photo of a prison cell with a window and a view of the ocean, and the word 'freedom' painted on the glass |
| 7    vehicle flying through a cyberpunk city 4 k, hyper detailed photograph |
| 8    a scene with a city in the background, and a single cloud in the foreground, with the text 'contemplate the clouds' in rounded cursive |
| 9    A pencil made of a tree branch with leaves |
| 10   A wooden table that has a silver trophy in the middle of it. In front of the trophy are several bowls and dishes containing food. There is a loaf of bread on a block of wood at the front of the table. |

Table 15: Selected prompts with a high level of disagreement in scores among raters for Gecko(R).

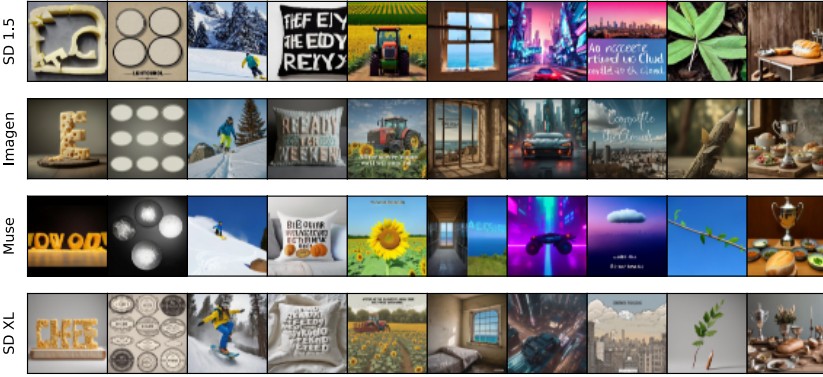

Figure 18: Images generated from Gecko(R) prompts with high level of disagreement among raters. Prompts are listed in Table 15.

| | High Disagreement Prompts (Gecko(S)) |
|---|---|
| 1 | the amazing view from the Halley Research Station in Antarctica on a clear night, the full Moon is rising and the sky is ablaze with the aurora australis, or polar lights. |
| 2 | a futuristic sculpture made of smooth metal |
| 3 | time lapse of sunrise over at the Hoover Dam |
| 4 | A huge vase in the middle of a field towering over the lawn chairs. |
| 5 | a bottle of Irn-Bru is sitting on a shelf. |
| 6 | a lord howe island palm tree with a moon rising in the distance |
| 7 | a long exposure image of the golden dunes at Playa del Ingles on the Canary Island, with a lone tourist |
| 8 | The soup is behind the cheese platter, to the left of the wine glasses, and below the crackers. |
| 9 | An alpaca and Chewbacca pose for a selfie at Machu Picchu |
| 10 | the lion cub named Simba is catching a ball. |

Table 16: Selected prompts with a high level of disagreement in scores among raters for Gecko(S).

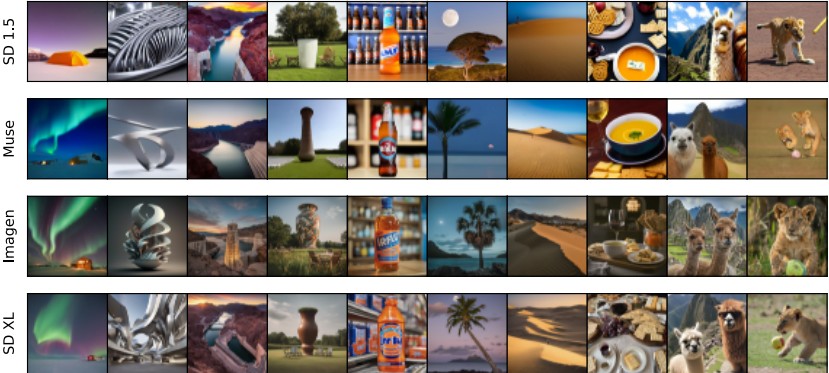

Figure 19: Images generated from Gecko(S) prompts with high levels of disagreement among raters. Prompts are listed in Table 16.

## D.4 HUMAN EVALUATION TEMPLATES: CHALLENGING CASES

In Figs. 20, 21, and 22 we show examples of challenging cases for the absolute comparison templates.

| Image | Ratings |
|---|---|
| | A Nexus One is placed on a bench. |
| | A Nexus One is placed on a bench. |
| | A Nexus One is placed on a bench. |
| | Some shirts and some pizzas. There are more shirts than pizzas. |
| | Some shirts and some pizzas. There are more shirts than pizzas. |
| | Some shirts and some pizzas. There are more shirts than pizzas. |

Figure 20: **Examples of challenges for WL.** We show two examples of evaluated images, respective prompts and annotations from three raters. Each word is coloured according to the score given by the rater: green indicates *Aligned*, red *Not aligned*, and yellow *Unsure*. Both examples show that WL can be sensitive to words that are not relevant to the alignment evaluation. **Top:** All raters seem to agree it is not possible to tell whether a bench is represented in the image (hence the word is evaluated as *Unsure*). In spite of that, one of the raters disagrees on how to rate the "on a" preposition. **Bottom:** All raters seem to agree the quantity of shirts in the image does not reflect the prompt, but their ratings vary in terms of which words are rated as not aligned.

| Image | Prompt | Rater 1 | Rater 2 | Rater 3 |
|---|---|---|---|---|
| | A giraffe stands in the field. | 4-Mostly consistent | 5-Consistent | 5-Consistent |
| | The raccoon holds the cat. | 2-Mostly inconsistent | 3-Somewhat consistent | 4-Mostly consistent |

Figure 21: **Examples of challenges for Likert. Top:** Raters might take into account other aspects of the images besides alignment when evaluating a prompt-image pair. In this example, although the image is perfectly consistent with the prompt, one of the raters penalised its score. We hypothesise they took into account the fact the generated image is in grey scale. **Bottom:** "Uncalibrated" scores across raters. The scores of all three raters reflect the imperfect consistency between prompt and image, but each rater penalised the score with different intensity.

| Image | Prompt | Questions |
|---|---|---|
| | A church without a steeple. | Is there a church? |
| | | Does the church have a steeple? |
| | | Is the steeple missing? |
| | A wood carving of an owl. | Is there an owl? |
| | | Is there a wood carving? |
| | | Is the wood carving made of wood? |

Figure 22: **Examples of challenges for DSG(H). Top:** Language complexity–Negation. As also shown in Fig. 4, the question generation is confused by the negation (asking if the church *has* a steeple as opposed to *does not have* a steeple). **Bottom:** Coverage. The question generation fails to capture that the owl should be represented as a wood carving.

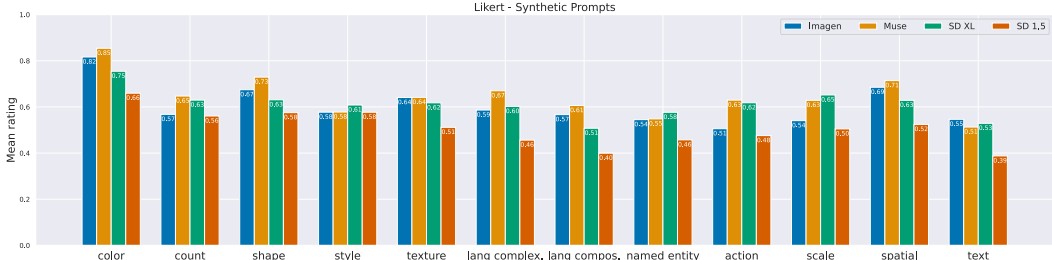

Figure 23: **Per skill results - Likert.** Muse scores the best in nine out of the twelve categories, and SD1.5 performs the worst in all categories. Focusing on Muse, SDXL, and Imagen, the models score above 0.5 on all categories. Recalling that the Likert scale is symmetric (0.0 being inconsistent, and 1.0 being consistent), we see that these three models are more consistent than inconsistent on average (albeit only slightly for skills such as 'lang compos.', 'named entity' and 'text').

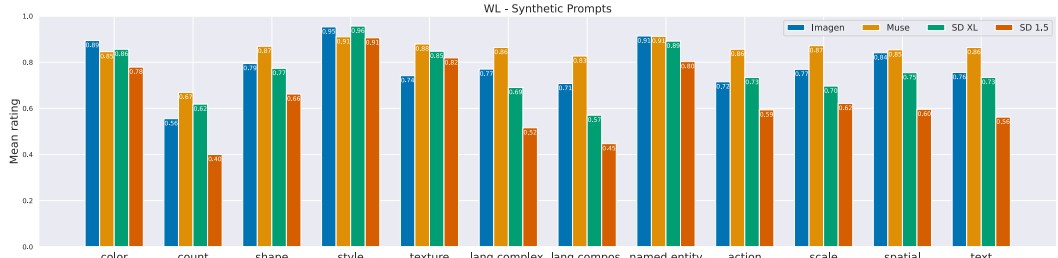

Figure 24: **Per skill results - WL.** Muse scores the best in ten out of the twelve skills, and SD1.5 performs the worst in all skills. Moreover, Muse scores higher than the other models by a noticeable margin ($\geq 0.1$) for the skills 'lang compos.', 'action', 'scale' and 'text'. In this case, analysing the results by skill shows that we can contribute Muse's higher average score (over the whole prompt set) mostly to these skills.

# E  T2I MODELS: ADDITIONAL COMPARISONS

## E.1  ANALYSING MODEL RATINGS PER SKILL

In Figures 23, 24 and 25, we plot the mean ratings in different skills for Likert, WL and DSG(H), respectively. We focus on Gecko(S) because we have a skill/sub-skill label for each prompt. Our goal is to understand how the trends in model performance on the whole prompt set relate to their performance in individual skills. We provide an overview of the results in the captions for the plots. Overall, the results broken down by skill are consistent with the averages over the whole prompt set. In other words, if a model is better or worse on the full prompt set, this is generally true for the individual categories as well. Another observation is that COUNTING and COMPLEX LANGUAGE seem to be the most difficult skills judging by WL and DSG, but this is not as clear from Likert (where many categories seem just as difficult).

**Further Breaking Down Skills.**   We can gain more insight into the skills of the models by looking at variation within a skill. Figure 26 shows sub-skills of the COLOUR prompts. We find that two sub-skills are more challenging, corresponding to prompts that require the models to combine multiple skills when generating the image (i.e., colour plus either composition or text rendering). For example, the 'colour:composed' sub-skill (COMPOSED EXPRESSIONS) includes prompts such as *'A brown vase, a white plate, and a red fork.'* with variations in the colors/objects. The sub-skill 'color:stroop' (STROOP) contains prompts like *'Text saying "green" in white letters.'* where the word in quotes differs from the color of the letters.

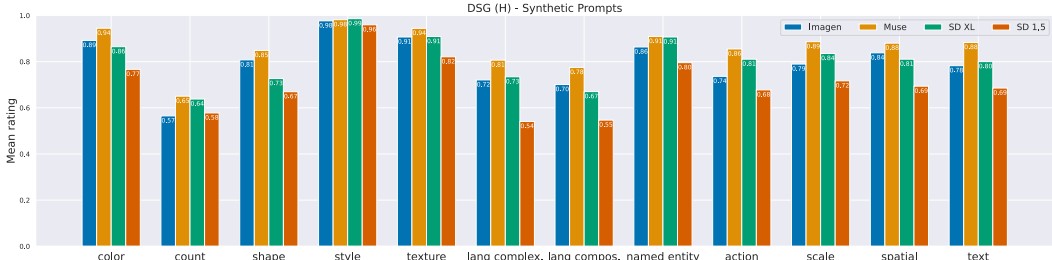

Figure 25: **Per skill results - DSG(H).** Muse performs well across the skills, being the best eleven out of twelve times (scoring very close to the top for 'style'). On the other hand, SD1.5 scores the worst in all the skills. This is consistent with the average scores on the overall prompt set. We see that counting is the most difficult skill for Muse, SDXL, and Imagen. Aside from counting, the hardest skills for Muse are the language ones ('lang complex.' and 'lang compos.'). This relative skill deficiency is not evident from the Likert and WL ratings, and therefore, the DSG ratings are better able to capture model shortcomings for prompts with more complex linguistic structure.

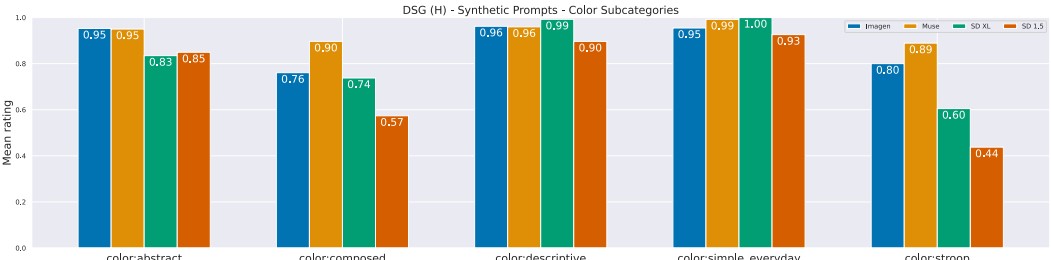

Figure 26: **Color sub-skill results - DSG (H).** We further break down the prompts in the color skill into five sub-skills. One observations is that two of the sub-skills ('composed' and 'stroop') are noticeably more difficult for all the models. This can be explained by the fact that they combine multiple skills: 'composed' includes prompts with multiple colors/objects, and 'stroop' includes text rendering in a certain color. Hence, while models may perform well on a skill overall, the sub-skills can illuminate where they struggle in generating images aligned with more complex prompts. On the other hand, models perform well with both abstract and everyday colors, likely because these are more commonly seen during training.

## E.2 Model comparisons with TIFA160

We augment the experiment from Fig. 13 in Sec. 4.2 by performing a similar analysis with TIFA160. We generate images with this set of prompts and perform human evaluation following the same protocol as for Gecko2K and its subsets. In Fig. 27, we show the results of pairwise model comparisons with TIFA160 carried out with the Wilcoxon signed-rank test with $p < 0.001$. Results show that all three versions of Gecko prompts are able to better distinguish models by finding more significant comparisons between them.

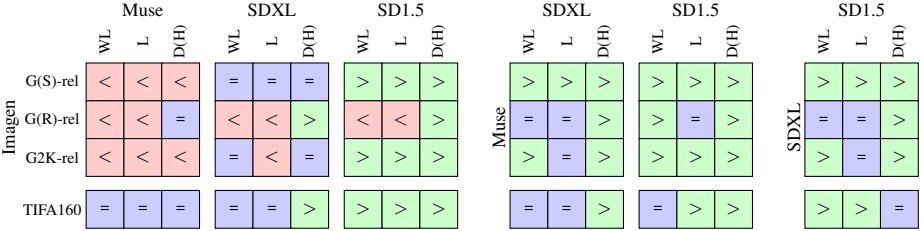

Figure 27: **Comparing models using human annotations.** We compare model rankings on the reliable subsets of Gecko(S) (G(S)-rel), Gecko(R) (G(R)-rel), both subsets (G2K-rel), and TIFA160. We perform the Wilcoxon signed-rank test for all pairs of models (p<0.001) and post-hoc comparison based on average ratings. Each grid represents a comparison between two models. Entries in the grid depict results for WL, Likert (L), and DSG(H) (D(H)) scores. The $>$ sign indicates the left-side model is better, worse ($<$), or not significantly different ($=$) than the model on the top. All three versions of Gecko prompts are able to better distinguish models by finding more significant comparisons between them.

# F Auto-eval metrics: Additional experiments on T2I

## F.1 Intuitive explanation of each task

We use this section to give an intuitive explanation of each task – *point-wise instance scoring, pairwise instance scoring, model ordering* – as well as how they can lead to different outcomes in terms of metric ranking on a toy setting. We evaluate the different auto-eval metrics on our actual dataset in Sec. 6, where we find that *in practice* metrics do achieve different rankings on these tasks.

**Point-wise instance scoring.** Point-wise instance scoring evaluates how well a metric ranks generations for a single model. Assume we have a set of generations for Model $A$ over a prompt set $\mathcal{P}$ and metric $m$ that calculates scores $m(A(p), p)$ for a generation $A(p)$ for a prompt $p \in \mathcal{P}$. We also have a human rating $h(A(p), p)$ on some numerical scale between $[0, 1]$ where 1 indicates a perfect generation and 0 a terrible one. This evaluation compares how well the metric scores correlate with the human ones over that prompt set for Model $A$. If we have multiple models, we average the correlation coefficients obtained for each model.

**Pair-wise instance scoring.** Pair-wise instance scoring compares two metrics given a pair of generations for a prompt. Assume we have two generations $A(p), B(p)$ for the prompt $p$ for Model $A$ and Model $B$. We also have a human preference (e.g. Model $A >$ Model $B$), and a metric $m$ which gives a score (e.g. $m(A(p), p)$ ) for a generation. Pairwise instance scoring would evaluate whether the relationship between the scores (i.e. $m(A(p), p) > m(B(p), p)$) matches that of the human rating. If the relationship matches, then we say $m$ is correct for that example, else it is incorrect. To get an average accuracy for a given metric within a dataset, we count the number of examples for which $m$ predicts the correct relationship for all prompts / model pairs and divide by the number of comparisons.

**Model ordering.** For model ordering, we compare two metrics over a set of prompts and corresponding generations. We average the scores for a metric across all prompts to get $m_{avg}(A) = \sum_p m(A(p), p)/|\mathcal{P}|$. We then evaluate (for all model pairs) how often the model ordering from human evaluation matches that obtained when comparing a given model pair: e.g. $m_{avg}(A)$ vs $m_{avg}(B)$ for Model $A$ and Model $B$.

| Model 1 | Model 2 | 'GT' | | Gecko | DSG | TIFA | CLIP | VNLI | VQAScore |
|---------|---------|------|---|-------|-----|------|------|------|----------|
| Imagen | Muse | < | | < | < | < | < | < | < |
| Imagen | SDXL | —— | | < | < | —— | < | —— | < |
| Imagen | SD1.5 | > | | > | > | > | < | > | > |
| Muse | SDXL | > | | > | —— | —— | < | —— | —— |
| Muse | SD1.5 | > | | > | > | > | > | > | > |
| SDXL | SD1.5 | > | | > | > | > | > | > | > |

Figure 28: **Comparing model ordering obtained from humans and auto-eval metrics on G2K-rel.** We show the 'GT' human ordering and the predicted ones for auto-eval metrics. $<$ means Model 1 $<$ Model 2, $>$ Model 1 $>$ Model 2 and $--$ that no significant relation was found. While CLIP performs poorly, mistaking wins with losses, no other metric confuses a win with a loss.

**Toy example.** Why these different evaluation procedures can give different results for a metric is subtle. Consider the following toy setting. We have two models: $A$ and $B$, and 3 prompts. For prompt $p_1$, Model $A$ is clearly much better than Model $B$, which is terrible, but for prompts $p_2, p_3$, Model $A$ is ever so slightly worse but both are reasonable. Intuitively, Model $A$ is better than Model $B$ on this prompt set.

1. *Example 1*: Now imagine we have a metric that can do a good job of doing pairwise instance scoring but is not well calibrated across prompts (e.g. a score does not indicate an overall notion of alignment – 0.7 for one prompt could correspond to a poor generation but a score of 0.5 on another denotes a high quality generation). So in this toy example, that metric gives the following scores for Model $A$: $p_1 = 0.1, p_2 = 0.1, p_3 = 0.1$ and for Model $B$: $p_1 = 0.05, p_2 = 0.8, p_3 = 0.8$. The comparisons are all right, but the average score for Model $A$ is $0.1$ and Model $B$ is $0.55$, which does not match the intuition that Model $A$ is actually better than Model $B$.

2. *Example 2*: We can conversely have a metric that is well calibrated across prompts but not able to reliably pick up on subtle differences. Such a metric could have the following scores for Model $A$: $p_1 = 0.8, p_2 = 0.7, p_3 = 0.72$ whereas Model $B$: $p_1 = 0.1, p_2 = 0.69, p_3 = 0.71$. While the model ordering is correct, the pairwise comparisons for prompts $p_2, p_3$ are not.

These examples demonstrate how a metric can be good at pairwise comparison (e.g. a metric like CLIP) but be poor at model ordering, i.e. a metric can give scores that are not well calibrated across prompts. Similarly, a metric can be good at model ordering but bad at pairwise instance scoring because it does not capture subtle differences. We can use a similar logic to understand why the point-wise instance scoring and pair-wise instance scoring tasks can achieve differing results as well as the model ordering and point-wise instance scoring tasks.

F.2   MODEL-ORDERING EVALUATION

We provide detailed results for the model ordering experiment. The goal is to determine the each auto-eval metric can predict the significant relations found from human annotation. We use G2K-rel as it is the largest subset with agreement among annotation templates across models. We take the majority vote to determine the relationship and note that there is no template that disagrees with this vote but one template may find a significant relation where the others do not, or vice versa. For each model pair, we compare distributions of auto-eval metric predictions using the same Wilcoxon signed rank test to get the relationship predicted by the metric. We plot in Table 28 the results. CLIP performs poorly, confusing wins with ties. However, no other auto-eval metric confuses a win with a tie showing that these auto-eval metrics are already robust for this task. Of these metrics, we see that TIFA and Gecko are able to get the most number of significant relationships right.

| Metrics | FT | Gecko(R) | | | | Gecko(S) | | | | Gecko(R)-Rel | | | | Gecko(S)-Rel | | | |
|---|---|---|---|---|---|---|---|---|---|---|---|---|---|---|---|---|---|
| | | WL | Likert | DSG(H) | SxS | WL | Likert | DSG(H) | SxS | WL | Likert | DSG(H) | SxS | WL | Likert | DSG(H) | SxS |
| | | SpearmanR | | | Acc | SpearmanR | | | Acc | SpearmanR | | | Acc | SpearmanR | | | Acc |
| *Interpretable (QA/VQA)* | | | | | | | | | | | | | | | | | |
| TIFA$_{\text{PALM-2/PALI}}$ | ✗ | 0.26 | 0.34 | 0.28 | 41.7 | 0.39 | 0.32 | 0.39 | 53.2 | 0.34 | 0.37 | 0.33 | 47.3 | 0.41 | 0.36 | 0.40 | 52.8 |
| DSG$_{\text{PALM-2/PALI}}$ | ✗ | 0.35 | 0.47 | 0.42 | 49.6 | 0.45 | 0.45 | 0.45 | 58.1 | 0.47 | 0.50 | 0.50 | 53.9 | 0.48 | 0.47 | 0.48 | 56.9 |
| Gecko$_{\text{PALM-2/PALI}}$ | ✗ | 0.41 | 0.55 | 0.46 | 62.1 | 0.47 | 0.52 | 0.45 | 74.6 | 0.52 | 0.58 | 0.53 | 71.3 | 0.52 | 0.54 | 0.49 | 75.2 |
| Gecko$_{\text{Gemini Flash}}$ | ✗ | **0.43** | **0.58** | **0.48** | **72.2** | **0.54** | **0.59** | **0.56** | **78.8** | **0.56** | **0.62** | **0.58** | **74.0** | **0.57** | **0.63** | **0.57** | **80.0** |
| *Uninterpretable (single score)* | | | | | | | | | | | | | | | | | |
| CLIP | ✗ | 0.14 | 0.16 | 0.13 | 54.4 | 0.25 | 0.18 | 0.26 | 67.2 | 0.11 | 0.09 | 0.08 | 59.7 | 0.24 | 0.19 | 0.25 | 67.1 |
| PyramidCLIP | ✗ | 0.26 | 0.27 | 0.26 | 64.3 | 0.22 | 0.25 | 0.23 | 70.7 | 0.26 | 0.26 | 0.23 | 65.8 | 0.21 | 0.26 | 0.22 | 71.0 |
| VQAScore$_{\text{Gemini Flash}}$ | ✗ | 0.42 | 0.54 | 0.45 | 73.1 | 0.51 | 0.57 | 0.49 | 76.5 | 0.51 | 0.59 | 0.52 | 73.9 | 0.54 | 0.60 | 0.51 | 77.0 |
| VNLI | ✓ | 0.37 | 0.49 | 0.42 | 54.4 | 0.45 | 0.55 | 0.45 | 72.7 | 0.49 | 0.57 | 0.46 | 65.6 | 0.50 | 0.61 | 0.48 | 72.7 |
| Gecko+VQAScore$_{\text{Gemini Flash}}$ | ✗ | – | – | – | 81.0 | – | – | – | 86.3 | – | – | – | 82.7 | – | – | – | 87.4 |

Table 17: **Correlation between VQA-based, contrastive, and fine-tuned (FT) auto-eval metrics and human ratings across annotation templates on Gecko2K and Gecko2K-Rel.** We observe a similar trend in Gecko2k and Gecko2K-Rel: Gecko performs the best across the board, and it can be improved by using a better language/VQA backend.

| Metrics | FT | Gecko(R) | | | | Gecko(S) | | | | Gecko(R)-Rel | | | | Gecko(S)-Rel | | | |
|---|---|---|---|---|---|---|---|---|---|---|---|---|---|---|---|---|---|
| | | WL | Likert | DSG(H) | SxS | WL | Likert | DSG(H) | SxS | WL | Likert | DSG(H) | SxS | WL | Likert | DSG(H) | SxS |
| | | Pearson | | | Acc | Pearson | | | Acc | Pearson | | | Acc | Pearson | | | Acc |
| *Interpretable (QA/VQA)* | | | | | | | | | | | | | | | | | |
| TIFA$_{\text{PALM-2/PALI}}$ | ✗ | 0.21 | 0.32 | 0.25 | 41.7 | 0.39 | 0.32 | 0.39 | 53.2 | 0.27 | 0.35 | 0.27 | 47.3 | 0.43 | 0.35 | 0.41 | 52.8 |
| DSG$_{\text{PALM-2/PALI}}$ | ✗ | 0.28 | 0.41 | 0.38 | 49.6 | 0.43 | 0.42 | 0.44 | 58.1 | 0.39 | 0.43 | 0.44 | 53.9 | 0.45 | 0.43 | 0.46 | 56.9 |
| Gecko$_{\text{PALM-2/PALI}}$ | ✗ | 0.38 | 0.51 | 0.42 | 62.1 | 0.46 | 0.48 | 0.46 | 74.6 | 0.50 | 0.55 | 0.51 | 71.3 | 0.52 | 0.52 | 0.50 | 75.2 |
| Gecko$_{\text{Gemini Flash}}$ | ✗ | **0.40** | **0.53** | **0.47** | **72.7** | **0.52** | **0.58** | **0.56** | **77.9** | **0.53** | **0.57** | **0.54** | **74.0** | **0.56** | **0.61** | **0.56** | **80.0** |
| *Uninterpretable (single score)* | | | | | | | | | | | | | | | | | |
| CLIP | ✗ | 0.15 | 0.18 | 0.16 | 54.4 | 0.26 | 0.19 | 0.25 | 67.2 | 0.13 | 0.12 | 0.10 | 59.7 | 0.26 | 0.19 | 0.25 | 67.1 |
| PyramidCLIP | ✗ | 0.29 | 0.30 | 0.26 | 64.3 | 0.28 | 0.27 | 0.25 | 70.7 | 0.31 | 0.28 | 0.25 | 65.8 | 0.28 | 0.28 | 0.26 | 71.0 |
| VQAScore$_{\text{Gemini Flash}}$ | ✗ | 0.35 | 0.44 | 0.36 | 73.1 | **0.42** | 0.53 | **0.41** | 76.5 | 0.41 | 0.47 | 0.42 | 73.9 | 0.46 | 0.56 | 0.42 | 77.0 |
| VNLI | ✓ | 0.34 | **0.48** | **0.39** | 54.4 | 0.41 | **0.55** | 0.42 | 72.7 | 0.25 | 0.41 | 0.22 | 65.6 | 0.35 | 0.49 | 0.34 | 72.7 |
| Gecko+VQAScore$_{\text{Gemini Flash}}$ | ✗ | – | – | – | 81.0 | – | – | – | 86.3 | – | – | – | 82.7 | – | – | – | 87.4 |

Table 18: **Pearson correlation between VQA-based, contrastive, and fine-tuned (FT) auto-eval metrics and human ratings across annotation templates on Gecko2K and Gecko2K-Rel.** Similar to the comparisons on Spearman correlation, Gecko again outperforms other auto-eval metrics (both QA/VQA and single score) with higher overall Pearson correlation. Swapping for the GeminiFlash backend leads to consistent performance improvement across templates.

## F.3 ADDITIONAL PAIR-WISE INSTANCE SCORING AND POINT-WISE INSTANCE SCORING RESULTS ON GECKO2K, GECKO-REL

We report the Spearman Rank correlation of different auto-eval metrics on Gecko2K in Table 4. Here we report the Pearson correlation in Table 18 as well and both Spearman and Pearson on Gecko-Rel in Table 17. Results follow those in the paper: the Gecko metric is consistently best though VQAScore is a strong baseline. While DSG/TIFA perform better than CLIP on the absolute templates, CLIP performs better on SxS.

In SxS comparison, to investigate how the interpretable and uninterpretable metrics can be combined to achieve better results, we also take the samples on which Gecko and VQAScore agree, and compute the accuracy of prediction on them. We found that we can improve agreement to >80% for this subset.

## F.4 RESULTS FOR ADDITIONAL CLIP METRICS ON THE GECKO BENCHMARK

We compare the Gecko metric with several score-based auto-eval metrics (Li et al., 2023; Ilharco et al., 2021; Yu et al., 2022a; Sun et al., 2023; Zhai et al., 2023; Gao et al., 2022; Zeng et al., 2022), as well as QA-based metrics such as TIFA (Hu et al., 2023) and DSG (Cho et al., 2023a), and VNLI models (Yarom et al., 2024) in Table 19.

Generally, our Gecko metric outperforms the others and shows a higher correlation with most of the human annotation templates. DSG is the second best metric, except on SxS where it ranks third. It outperforms TIFA by a clear margin but falls behind Gecko. Finally, we note that Gecko even shows

| Metrics | Gecko (R) | | | | | | Gecko (S) | | | | | |
|---|---|---|---|---|---|---|---|---|---|---|---|---|
| | WL | | Likert | | DSG(H) | | WL | | Likert | | DSG(H) | |
| | Pearson | Spearman-R | Pearson | Spearman-R | Pearson | Spearman-R | Pearson | Spearman-R | Pearson | Spearman-R | Pearson | Spearman-R |
| BLIP-2$_{\text{ITM}}$ | 0.25 | 0.22 | 0.24 | 0.19 | 0.23 | 0.21 | 0.28 | 0.23 | 0.13 | 0.16 | 0.25 | 0.23 |
| CLIP-B/32 | 0.15 | 0.14 | 0.18 | 0.16 | 0.16 | 0.13 | 0.26 | 0.25 | 0.19 | 0.18 | 0.25 | 0.26 |
| CLIP-B/32$_{\text{LAION-2B}}$ | 0.26 | 0.21 | 0.24 | 0.22 | 0.26 | 0.21 | 0.28 | 0.24 | 0.23 | 0.22 | 0.26 | 0.24 |
| CLIP-B/16 | 0.16 | 0.11 | 0.15 | 0.12 | 0.17 | 0.12 | 0.26 | 0.22 | 0.15 | 0.14 | 0.24 | 0.22 |
| CLIP-L/14 | 0.18 | 0.16 | 0.16 | 0.14 | 0.18 | 0.15 | 0.26 | 0.23 | 0.17 | 0.16 | 0.25 | 0.24 |
| CLIP-H/14$_{\text{LAION-2B}}$ | 0.29 | 0.24 | 0.25 | 0.22 | 0.27 | 0.23 | 0.29 | 0.24 | 0.23 | 0.22 | 0.27 | 0.25 |
| CLIP-g/14$_{\text{LAION-2B}}$ | 0.30 | 0.24 | 0.26 | 0.23 | 0.28 | 0.23 | 0.30 | 0.25 | 0.24 | 0.23 | 0.28 | 0.25 |
| CLIP-G/14$_{\text{LAION-2B}}$ | 0.29 | 0.23 | 0.25 | 0.21 | 0.27 | 0.23 | 0.30 | 0.24 | 0.25 | 0.23 | 0.28 | 0.25 |
| CoCa-L/14 | 0.28 | 0.23 | 0.26 | 0.22 | 0.26 | 0.21 | 0.29 | 0.25 | 0.24 | 0.22 | 0.28 | 0.26 |
| EVA-02-CLIP-L/14 | 0.27 | 0.24 | 0.23 | 0.21 | 0.24 | 0.22 | 0.30 | 0.26 | 0.21 | 0.20 | 0.28 | 0.27 |
| EVA-02-CLIP-E/14 | 0.28 | 0.23 | 0.24 | 0.20 | 0.27 | 0.22 | 0.28 | 0.23 | 0.23 | 0.22 | 0.26 | 0.24 |
| EVA-02-CLIP-E/14+ | 0.30 | 0.24 | 0.24 | 0.21 | 0.27 | 0.23 | 0.29 | 0.24 | 0.25 | 0.23 | 0.28 | 0.25 |
| SigLIP-B/16 | 0.26 | 0.21 | 0.22 | 0.18 | 0.27 | 0.21 | 0.29 | 0.25 | 0.22 | 0.21 | 0.29 | 0.26 |
| SigLIP-L/16 | 0.28 | 0.24 | 0.26 | 0.22 | 0.29 | 0.25 | 0.29 | 0.26 | 0.23 | 0.22 | 0.29 | 0.27 |
| PyramidCLIP-B/16 | 0.29 | 0.26 | 0.30 | 0.27 | 0.29 | 0.26 | 0.28 | 0.22 | 0.27 | 0.25 | 0.25 | 0.23 |
| X-VLM$_{\text{16M}}$ | 0.17 | 0.11 | 0.21 | 0.09 | 0.25 | 0.15 | 0.26 | 0.23 | 0.23 | 0.16 | 0.24 | 0.23 |
| TIFA$_{\text{PALM-2/PALI}}$ | 0.21 | 0.26 | 0.32 | 0.34 | 0.25 | 0.28 | 0.39 | 0.39 | 0.32 | 0.32 | 0.39 | 0.39 |
| DSG$_{\text{PALM-2/PALI}}$ | 0.28 | 0.35 | 0.41 | 0.47 | 0.38 | 0.42 | 0.43 | 0.45 | 0.42 | 0.45 | 0.44 | 0.45 |
| Gecko$_{\text{PALM-2/PALI}}$ | 0.38 | 0.41 | 0.51 | 0.55 | 0.42 | 0.46 | 0.46 | 0.47 | 0.48 | 0.52 | 0.46 | 0.45 |
| Gecko$_{\text{Gemini Flash}}$ | **0.40** | **0.42** | **0.53** | **0.57** | **0.47** | **0.47** | **0.52** | **0.54** | **0.58** | **0.60** | **0.56** | **0.56** |
| VQAScore$_{\text{Gemini Flash}}$ | 0.35 | **0.42** | 0.44 | 0.54 | 0.36 | 0.45 | 0.42 | 0.51 | 0.53 | 0.57 | 0.41 | 0.49 |
| VNLI | 0.34 | 0.37 | 0.48 | 0.49 | 0.39 | 0.42 | 0.41 | 0.45 | 0.55 | 0.55 | 0.42 | 0.45 |

Table 19: **Correlation between auto-eval metrics and human ratings across three annotation templates on Gecko2K.** Best results per model type are underlined; best results are in **bold**.

higher correlation than the supervised VNLI model. By using a stronger, Gemini Flash backend, Gecko performs best by a significant margin consistently.

Looking at efficient, score-based metrics, we find that PyramidCLIP achieves competitive correlations. Moreover, a larger pre-training corpus leads to better metrics; e.g., as seen by comparing CLIP-B/32 (trained on 400M images) and CLIP-B/32$_{\text{LAION-2B}}$ (trained on 2B images). Finally, larger models are often better (e.g., SigLIP-L/16 vs. SigLIP-B/16), although these trends are less consistent (e.g., EVA-02-CLIP-L/14+ ≈ EVA-02-CLIP-E/14).

## F.5 ANALYSING AUTO-EVAL METRIC RESULTS PER SKILL.

We present the per-skill Spearman Ranked Correlation between different auto-eval metrics and human annotation templates in Fig. 29. We observe a similar trend across the three plots, as we discussed in Sec. 6: Gecko is the best on handling prompts with "language complexity", which can be attributed to the coverage tagging and filtering steps in its pipeline that make Gecko less prone to errors when processing long and complicated prompts. DSG is better on "compositional prompts", as it can leverage its utilization of dependency graphs. VNLI and VQAScore demonstrate advantages in assessing "shape", "color", and "text" (e.g., TEXT RENDERING) prompts though we note they are both worse on the more complex prompts (e.g., "compositional" and "language complexity"). When leveraging a better QA/VQA model (e.g., GeminiFlash) for Gecko, we see improvements across the board; Gecko with GeminiFlash performs consistently the same or better than VNLI/VQAScore. TIFA and CLIP consistently perform poorly and worse than the other metrics. It is also worth noting that all metrics exhibit relatively poor performance on "text", "style", and "named identity", highlighting the current lack of OCR and named recognition ability in existing contrastive, NLI and VQA models.

## F.6 ADDITIONAL VISUALISATIONS.

We visualise additional examples in Fig. 30 for different categories (spatial, counting, text rendering, linguistic complexity, etc.). These examples demonstrate both differences arising from different annotation templates and also different metrics.

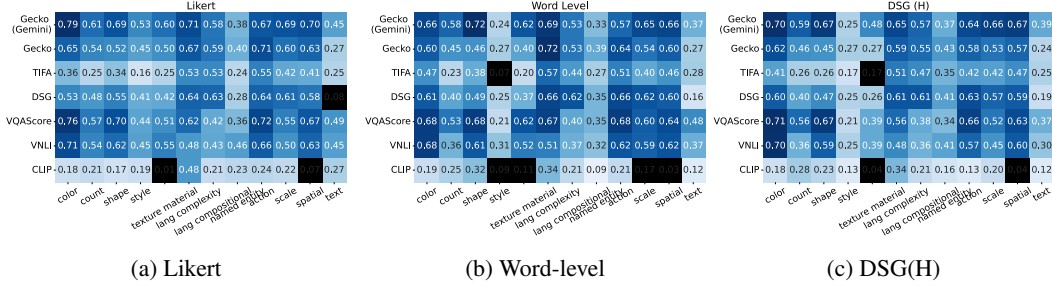

| (a) Likert | (b) Word-level | (c) DSG(H) |

Figure 29: **Per skill results by metric.** We visualise the correlation for each skill. Where p-values are > 0.05, we color the square black.

| Prompt: | A snake is on the elephant. | | | | a bird flying over the mountains in the sunset, with the text "bla bla bla, this is the sound of a helicopter" | | | |
|---|---|---|---|---|---|---|---|---|
| | Imagen | Muse | SDXL | SD1.5 | Imagen | Muse | SDXL | SD1.5 |
| APIC: | 1. | 1.0 | 0.44 | 0.33 | 0.53 | 0.83 | 0.47 | 0.21 |
| Likert: | 1. | 0.80 | 0.6 | 0.4 | 0.6 | 0.6 | 0.6 | 0.4 |
| DSG (H): | 1. | 0.50 | 1. | 0.5 | 0.92 | 0.8 | 0.71 | 0.25 |
| Gecko: | 0.98 | 0.84 | 0.73 | 0.26 | 0.85 | 0.58 | 0.86 | 0.21 |
| DSG: | 1.0 | 1.0 | 1.0 | 0.33 | 0.67 | 0.50 | 0.89 | 0.11 |
| VNLI: | 0.94 | 0.93 | 0.14 | 0.23 | 0.19 | 0.30 | 0.34 | 0.27 |

| Prompt: | There are 5 apples, two of them are yellow and two are black but none are red. | | | | the dog who wears a white shirt holds a beer | | | |
|---|---|---|---|---|---|---|---|---|
| | Imagen | Muse | SDXL | SD1.5 | Imagen | Muse | SDXL | SD1.5 |
| APIC: | 0.88 | 0.73 | 0.57 | 0.61 | 1.0 | 1.0 | 1. | 0.4 |
| Likert: | 0.73 | 0.60 | 0.53 | 0.60 | 0.8 | 0.93 | 1. | 0.53 |
| DSG (H): | 0.80 | 0.53 | 0.5 | 0.40 | 0.75 | 1.0 | 1. | 0.25 |
| Gecko: | 0.88 | 0.91 | 0.84 | 0.69 | 0.98 | 0.97 | 0.97 | 0.7 |
| DSG: | 1.0 | 1.0 | 0.9 | 0.8 | 1. | 0.83 | 1. | 0.17 |
| VNLI: | 0.23 | 0.20 | 0.19 | 0.16 | 0.9 | 0.95 | 0.94 | 0.17 |

| Prompt: | A fortune cookie that has the fortune "the best way to predict the future is to create it." | | | | A brown glass salad bowl on a grey metal table. | | | |
|---|---|---|---|---|---|---|---|---|
| | Imagen | Muse | SDXL | SD1.5 | Imagen | Muse | SDXL | SD1.5 |
| APIC: | 0.35 | 0.39 | 0. | 0.02 | 1.0 | 0.88 | 1.0 | 0.63 |
| Likert: | 0.53 | 0.47 | 0.4 | 0.27 | 0.73 | 0.67 | 0.73 | 0.53 |
| DSG(H): | 1.0 | 0.5 | 0.67 | 0.17 | .83 | 1.0 | 0.83 | 0.83 |
| Gecko: | 0.96 | 0.79 | 0.75 | 0.62 | 0.92 | 0.88 | 0.87 | 0.70 |
| DSG: | 0.88 | 0.25 | 1.0 | 0.25 | 0.93 | 0.93 | 0.86 | 0.79 |
| VNLI: | 0.37 | 0.27 | 0.25 | 0.35 | 0.81 | 0.65 | 0.55 | 0.31 |

Figure 30: **Additional visualisations of scores from different auto-eval metrics.** We show the image generations by the four generative models given two prompts from Gecko(S), with the alignment scores predicted by human annotators and auto-eval metrics respectively.

## F.7 RESULTS PER WORD FOR WL

Here we evaluate how well the Gecko metric (with the PALI/PALM-2 backend) can identify whether words are or are not grounded as rated by WL. This experiment can *only* be done for Gecko as *no other* metric gives word level annotations that can be traced back to the original words in the prompt. We note that this is much more challenging than giving an overall image rating. In order to perform this experiment, we first parse the coverage prediction to ensure we can match words in the original prompt with those in the coverage prediction. For example, if we have the original prompt 'a red-colored dog' and 'a {1}[red-colored] {2}[dog]' as the generated coverage one, we can map from the word index (e.g.'{1}') to the phrase 'red-colored'.

| | Gecko(R) | | | Gecko(S) | | |
|---|---|---|---|---|---|---|
| | # Words evaluated | Accuracy | Error Consistency ($\kappa$) | # Words evaluated | Accuracy | Error Consistency ($\kappa$) |
| SD1.5 | 3727 | 75.5 | 0.20 | 2729 | 77.5 | 0.52 |
| SDXL | 3901 | 80.0 | 0.13 | 3304 | 79.5 | 0.34 |
| Muse | 2792 | 82.5 | 0.24 | 3298 | 82.0 | 0.19 |
| Imagen | 2549 | 76.5 | 0.29 | 3472 | 80.2 | 0.39 |

Table 20: **Word level results comparing Gecko to the WL annotation template.** We can see that Gecko achieves high accuracy but also that results are not simply explained by chance, as $\kappa > 0$ and in general indicates fair agreement.

We then take all word level predictions where all annotators either annotated the word as grounded or not grounded (we removed those for which a subset of annotators annotated the word as 'unsure'). For these words, we take the ones where the coverage model predicts that it should be covered (e.g.in the example above, even if 'a' was always annotated as grounded, we would ignore it as it is not considered groundable by the coverage step). Given this final set of words, we look at whether the VQA prediction was accurate and compare this to whether the annotators thought that the word was grounded or not.

We report three numbers in Table 20: (1) the number of words we are left with, (2) the accuracy and (3) the error consistency $\kappa$ (Geirhos et al., 2020), equation (1),(3). We report error consistency as many of the words ($\sim 90\%$) are rated as grounded. Accuracy does not account for the fact that a metric which predicts 'grounded' $\sim 90\%$ of the time would actually get $\sim 80\%$ accuracy by chance. Error consistency takes this into account such that $\kappa = -1$ means that two sets of results never agree, $\kappa = 0$ that the overlap is explained by chance and $\kappa = 1$ means results agree perfectly. As shown by the results, Gecko is able to predict grounding at the word level with reasonable accuracy. Moreover, results are not simply explained by chance (as $\kappa > 0$.); the error consistency results indicate in general some but not substantial agreement.

# G EXTENDING GECKO TO MORE MODALITIES: TEXT-TO-VIDEO GENERATION

Video evaluations are more challenging, in that there are multiple aspects that are encapsulated in the text-to-video consistency, including stylistic, temporal, semantic and overall fidelity. In order to extend our approach, we follow a similar two step process, in which question-answer pairs are generated using a few-shot prompt that outlines which aspects of the video need to be covered by those pairs. The corresponding few-shot prompt used for question-answer pair generation with sufficient coverage of the groundable words in the prompt is shown in Listing 5.

```
1  """
2  Given a video description and the groundable words in it, generate multiple-choice questions that verify if
       the video description is correct.
3  The goal is to ask questions about entities, objects, attributes, actions, colors, spatial relations, temporal
        relations, styles and scenes, when these are present in the description.
4  Make sure that all options are substantially different from each other and only one option can be the correct
       one based on the description. Do not include other parts of the description as a non correct option.
5  Justify why the other options cannot be true based on the description and question. Also, make sure that the
       question cannot be answered correctly only based on common sense and without reading the description.
6  Each generated question should be independent of the other ones and it should be able to be understood without
       knowing the other questions; avoid refering to entities/objects/places from previous questions.
7  Finally, avoid asking very general questions, such as 'What is in the video?', or 'Name a character in the
       video'.
8  Generate the multiple-choice questions in the exact same format as the examples that follow. Do not add
       asterisks, white spaces, or any other reformatting and explanation that deviate from the formatting of
       the following examples.
9
10 Description:
11 A fat rabbit wearing a purple robe walking through a fantasy landscape.
12 The visual-groundable words and their scores are labelled below:
13 A {1}[fat, attribute, 1.0] {2}[rabbit, entity, 1.0] {3}[wearing a {4}[purple, color, 1.0] robe, attribute,
       1.0] {5}[walking, action, 1.0] through a {6}[fantasy landscape, scene, 1.0].
14 Generated questions and answers are below:
15 About {1}:
16 Q: What is the most appropriate description for the animal of the video?
17 Choices: thin, regular, slim, fat
18 A: fat
19 Justification: the rabbit in the video is fat ({1}). The options thin and slim are oposite of the attribute
       mentioned in the description and the regular adjective checks whether it is obvious that the rabbit has
       a weight above normal.
20 About {2}:
21 Q: Who wears a robe in the video?
22 Choices: rabbit, hare, squirrel, rat
```

```
23  A: rabbit
24  Justification: the rabbit is the animal that wears a robe in the video ({2}). Hare is an animal very similar
        to rabbit, and the other two options (squirrel and rat) are also similar but not true according to the
        description.
25  About {3}:
26  Q: What is the rabbit wearing in the video?
27  Choices: nothing, dress, robe, jumpsuit
28  A: robe
29  Justification: the rabbit is wearing a robe ({3}). Nothing is what normally an animal is wearing, and the
        options dress and jumpsuit are similar to the robe but not true according to the description.
30  About {4}:
31  Q: What is the color of the clothing that the rabbit wears in the video?
32  Choices: purple, blue, pink, green
33  A: purple
34  Justification: the rabbit is wearing a purple robe ({4}). the options blue, pink and green are colors similar
        to purple.
35  About {6}:
36  Q: What is the rabbit doing in the video?
37  Choices: running, walking, standing, jumping
38  A: walking
39  Justification: the rabbit is walking through a fantasy landscape ({5}, {6}). The options running and standing
        are similar to walking, and jumping is an action that could be performed by a rabbit, but not true
        according to the description.
40  About {7}:
41  Q: Where is the video taking place?
42  Choices: fields, countryside, fantasy landscape, mountains
43  A: fantasy landscape
44  Justification: the rabbit is walking through a fantasy landscape ({6}). The options fields, countryside, and
        mountains are different types of landscapes, but they are real-world scenes instead of fantasy ones.
45
46  Description:
47  A beautiful coastal beach in spring, waves lapping on sand by Hokusai, in the style of Ukiyo
48  The visual-groundable words and their scores are labelled below:
49  A {1}[beautiful coastal beach, scene, 1.0] {2}[in spring, temporal relation, 1.0], {3}[waves, scene, 1.0] {4}[
        lapping, action, 1.0] {5}[on sand, spatial relation, 1.0] {6}[by Hokusai, style, 1.0], {7}[in the style
        of Ukiyo, style, 1.0]
50  Generated questions and answers are below:
51  About {1}:
52  Q: Where is the video taking place?
53  Choices: cliffs, harbor, coastal park, coastal beach
54  A: coast beach
55  Justification: the main scene is a beautiful coastal beach ({1}). The options cliffs, harbor, and coastal park
         are similar to coastal beach but not true according to the description.
56  About {2}:
57  Q: Which season is most likely during the video?
58  Choices: spring, summer, autumn, winter
59  A: spring
60  Justification: the video shows a coastal beach in spring ({2}). The options summer, autumn and winter are
        other seasons that are not true according to the description.
61  About {3}:
62  Q: What is the level of movement of the sea during the video?
63  Choices: calm, wavy, slightly moving, ripply
64  A: wavy
65  Justification: the sea is wavy ({3}). The options calm, slightly moving, and ripply are different levels of
        movement of the sea and they are all different enough from wavy.
66  About {4}:
67  Q: What is the movement of the sea during the video?
68  Choices: gentle waves are coming to the shore, there is a tide, waves are lapping on the shore, there are sea
        ripples
69  A: waves are lapping on the shore
70  Justification: the sea is lapping on the shore ({4}). The other provided options are either of less intesity (
        gentle waves are coming to the shore, there are sea ripples) or the exact opposite (there is a tide).
71  About {5}:
72  Q: Where does the sea move to during the video?
73  Choices: sand, rocks, cliffs, pebbles
74  A: sand
75  Justification: the waves are lapping on sand ({5}). The options pebbles, rocks, and cliffs are different types
         of ground typically by the sea and have different levels of solidity.
76  About {6}:
77  Q: Whose artist is the theme of the scene similar to?
78  Choices: Utamaro, Hokusai, Hiroshige, Yoshitoshi
79  A: Hokusai
80  Justification: the theme of the scene resembles a painting of Hokusai. The other options are other Japanese
        artists that are similar to Hokusai.
81  About {7}:
82  Q: Which Japanese painting style is most similar to the video?
83  Choices: Ukiyo, Nihonga, Sumi, ink calligraphy
84  A: Ukiyo
85  Justification: the video scene is in the style of Ukiyo ({7}). The other options are other types of Japanese
        painting styles that are not similar to the video according to the description.
86
87  Description:
88  ...
89  """
```

Listing 5: Sample LLM template for generating QAs with coverage for videos.

**Evaluation setup.** As described in Section 6.5, we choose a prompt set that is appropriate for measuring alignment between the description and the video and a set of text-to-video models to collect human annotations using different templates. We consider a subset of 94 prompts from the

| Model 1 | Model 2 | | 'GT' | | Gecko | VideoCLIP | VQAScore |
|---------|---------|---|------|---|-------|-----------|----------|
| Lumiere | Phenaki | | > | | > | −− | −− |
| Lumiere | WALT | | > | | > | −− | > |
| Phenaki | WALT | | −− | | −− | −− | > |

Figure 31: **Comparing model ordering obtained from humans and auto-eval metrics on VBench.** We show the 'GT' human ordering and the predicted ones for auto-eval metrics. < means Model 1 < Model 2, > Model 1 > Model 2 and −− that no significant relation was found.

VBench benchmark (Huang et al., 2024b) manually tagged with "overall consistency" for evaluating overall text-to-video alignment by the curators of the benchmark. We compare the following text-to-video models: Lumiere (Bar-Tal et al., 2024), Phenaki (Villegas et al., 2022) and WALT (Gupta et al., 2023). For human evaluation, we consider both absolute comparison templates (i.e., Likert and Word Level) and a template for relative pairwise comparisons (i.e., SxS), and for automatic evaluation, we again benchmark two types of auto-eval metrics: contrastive models (i.e., VideoCLIP; Xu et al. 2021) and VQA-based metrics. For VQA-based metrics, since there is no prior work on text-to-video generation, we extend the VQAScore and our fine-grained Gecko metric on the video domain using Gemini Flash as our video question answering model, which can process long context multimodal inputs.

**Model ordering.** In addition to the Pair-wise instance scoringand point-wise instance scoring results presented in Table 7, we also compare model rankings per metric in Figure 31 using the Wilcoxon signed-rank test for all pairs of models. To obtain the ground-truth (GT) model ordering, we average human preferences across the three templates (Likert, WL, SxS) via majority voting and consider valid model rankings only when pairwise differences are significant ($p<0.05$). We find that the model ranking provided by Gecko agrees with the human rankings for all three model comparisons. In contrast, VideoCLIP, which is often used as part of tool use for text-to-video evaluation (Huang et al., 2024b; Liu et al., 2024), does not provide *any* information about the relative performance of video models in terms of alignment. VQAScore agrees with the ground truth comparisons only for one model pair (Lumiere vs WALT).