# OpenReview forum: "Revisiting text-to-image evaluation with Gecko: on metrics, prompts, and human rating"
_ICLR.cc/2025/Conference — ICLR 2025 Spotlight_

### Official Review · Reviewer_dzMN · 2024-11-03

**Soundness:** 4
**Presentation:** 3
**Contribution:** 3
**Rating:** 8
**Confidence:** 4

**Summary:**

The paper introduces a comprehensive benchmark for text-to-image evaluation. The main contribution is the Gecko dataset: which consists of 2k prompts (1k from existing DSG, and 1k fresh prompts), along with images from 4 T2I models (SD1.5, SDXL, Imagen Vermeer, and Muse). The key insight in the paper is that annotations from different formats (i.e likert scale/absolute rating, pairwise comparisons and DSG type scoring) can end up with different conclusions, and to that end, the paper suggests model ordering evaluations using all these metrics. Finally, the paper also presents Gecko, a evaluation metric based on Q/A with some fixes for improved Q/A generation, which seems to be a better alternative for evaluating T2I models.

**Strengths:**

The paper is quite rigorous in its methodology and seems to comprehensively validate several evaluation methodologies for T2I. In general, this might be one of the most statistically sound evaluations of T2I models and has a lot of insights for future work on this topic.
Further, the data provided in the paper (assuming public release) would also be useful to evaluate newer reward models/metrics in the future.

**Weaknesses:**

The biggest drawback/weakness that I can see in this paper is that it predominantly evaluates only 4 T2I models on 1/2k prompts (excluding the results with the gecko metric later on). While this already may be an improvement over prior work, I am curious as to would there be benefits to scaling the prompts 10x? Or are the results/conclusions at this level itself sufficient and there would be mostly diminishing returns from increasing the scale of the evaluations?

From the paper, it's not immediately clear whether all the data from the Gecko evaluation shall be released publicly. While the findings of the paper are valuable in itself, the contributions of the paper are significantly determined by the public release (or lack thereof) of the data.


The authors may also want to acknowledge a concurrent work on similar lines[1]


[1] Saxon et al. "Who Evaluates the Evaluations? Objectively Scoring Text-to-Image Prompt Coherence Metrics with T2IScoreScore (TS2)", NeurIPS 2024

**Questions:**

I would like to confirm that I understand correctly: In Fig. 2 an "=" in a grid cell means that for the considered evaluation protocol (e.g Likert, SxS), the difference in the performances of the model are not statistically significant (even if the absolute score is different?)

---

> ### Author Response · Authors · 2024-11-15
> **Response**
>
> We thank the reviewer for their time and appreciate that they valued the rigorous and comprehensive evaluation.
>
> We respond to their questions below:
>
> **RQ1: “The biggest drawback/weakness that I can see in this paper is that it predominantly evaluates only 4 T2I models on 1/2k prompts (excluding the results with the gecko metric later on). While this already may be an improvement over prior work, I am curious as to would there be benefits to scaling the prompts 10x? Or are the results/conclusions at this level itself sufficient and there would be mostly diminishing returns from increasing the scale of the evaluations?”**
>
> *Response:* This is a very interesting point. We note that our prompts were carefully curated so as to provide discriminability of models and metrics, therefore scaling to 10x prompts wouldn't be a trivial task; simply generating more prompts wouldn't necessarily provide more signal in terms of model differences if similar care is not taken. In summary, the size of a dataset is not meaningful if we don’t have a clear understanding of what skills that dataset is evaluating. We focussed on coverage, which meant prompt selection was more labour intensive, limiting its size.
>
> However, our results do indicate that the conclusions are generalizable. In Fig 13 (of the revision), when comparing G(S) to G(S)-rel model ordering results, we can see that the G(S)-rel subset gives the same conclusions and they both consistently find significant relationships. When comparing G(R) to G(R)-rel, we can see that there is less agreement originally but for the two model comparisons where templates do not disagree and there are many significant relationships across a prompt subset (e.g. Muse vs SD1.5 and SDXL vs SD1.5) then both  G(R) and G(R)-rel give the same conclusion. We also have results on the Tifa160 prompt set (which is much smaller, with only 160 prompts) in Appendix E.2. Here, we find that it obtains a similar model ordering to G2K-rel but with many fewer significant relationships.
> These results indicate that if we get consistent, significant results for model comparisons across annotation templates and the prompt set is of a similar distribution, then results generalise to a larger set. If we get inconsistent results or insignificant results across annotation templates then a larger prompt set would potentially help to disentangle those subtle differences.
>
> For the metric comparisons, we note that we evaluated across our dataset but the conclusions around our metric’s superior performance generalised to Tifa160 (Table 6) and to a video benchmark (Table 7), demonstrating these results do generalise as well.
>
> **RQ2: “From the paper, it's not immediately clear whether all the data from the Gecko evaluation shall be released publicly. While the findings of the paper are valuable in itself, the contributions of the paper are significantly determined by the public release (or lack thereof) of the data.”**
>
> *Response:* We are in the process of open sourcing and have included extensive information in the paper so that the community can already duplicate our data collection and human annotation methodology in their own work. On the open sourcing front, we will release prompts, images and annotations subject to safety and privacy considerations (e.g. checking that there are no faces or potentially harmful material). On the methodological front, we include screenshots and extensive information in Appendix D of our human annotation framework. We also include extensive information in Section B around how we select prompts from existing datasets, tag and resample to give Gecko(R) as well as example templates we use to generate Gecko(S) including a full breakdown of skills and subskills with examples. Thus a reader has the information to duplicate the full annotation and data gathering process.
>
>
> **RQ3: “The authors may also want to acknowledge a concurrent work on similar lines[1].”**
>
> *Response:* Thank you for the suggestion. We have added the citation.
>
> **RQ4: “I would like to confirm that I understand correctly: In Fig. 2 an "=" in a grid cell means that for the considered evaluation protocol (e.g Likert, SxS), the difference in the performances of the model are not statistically significant (even if the absolute score is different?)”**
>
> *Response:* That is correct: ‘=’ denotes lack of statistical significance (but there may indeed be a difference in the absolute scores).

---

> > ### Comment · Reviewer_dzMN · 2024-11-25
> >
> > Thank you for your clarifications, I am happy to maintain my current (positive) rating.

---

### Official Review · Reviewer_GY7M · 2024-11-04

**Soundness:** 4
**Presentation:** 3
**Contribution:** 4
**Rating:** 8
**Confidence:** 5

**Summary:**

- Automatic evals are an effective alternative to human judgement to evaluate text-to-image (T2I) models.
- In this work authors perform a rigorous analysis on evaluation prompt sets, metrics and templates to derive reliable conclusions.
- Authors identify that different conclusions can be drawn when the prompt sets or evaluation templates are changed.
- This work introduces a new prompt set Gecko2K, a statistically grounded method to compare T2I models and a new framework to systematically evaluate metrics under model ordering, pair-wise instance scoring and point-wise instance scoring.
- Based on their analysis, authors propose a new automatic evaluation method that is highly correlated with human judgement across different human templates and evaluation settings.

**Strengths:**

- The authors conduct a rigorous study on the methodologies used for evaluating the text faithfulness of T2I models.
- The insights and takeaways from this paper are highly useful to advance research in both modeling and evaluating generative models.
- The experiments are very exhaustive and support all the claims made in the paper.
- The auto-eval method developed to increase coverage, remove hallucinations and normalize the VLM scores is novel and effective.
- A lot of thought and analysis has gone into designing the Gecko-2k reliable prompt set and future research in T2I can benefit from this work.
- Authors demonstrate that this method is not just limited to T2I models but can be easily extended to any modality generation models.

**Weaknesses:**

- The paper does not read well and requires multiple passes to properly assimilate.
- The terminology used is not intuitive and have not been used in prior literature making it harder to put them in context. Authors mention several terms like "skills", "reliable prompts" without properly introducing them creating confusion at several places. I would recommend authors first introduce the terms (what are skills? Maybe introduce them in the introduction.
Providing an example: a) The term "skills" was first introduced in L044 and the readers are directed to Table1 for more details. It has only become clear what those are after reading Table 8 (in the supplementary material). The column names in Table 8 are also different (Prompt Category and Subcategory and not skills and subskills). After finishing the introduction, the reader is still unsure what skills are. I have no issues with the term "skills" but authors can clarify what skills and subskills are in the first 2-3 paragraphs of the introduction(where the term was first introduced). Maybe L190-194 can be copied to the introduction so the readers know what to expect?

- A few suggestions, Fig. 2 is very hard to understand. Is there another way to convey the same information which is easier to read? There are too many rows making it hard to parse. Can you break the prompt sets to make it less busy? All the reliable prompt sets can be put in a separate figure and moved to the supplementary. From my understanding, the reliable prompts help reduce rater disagreement and not central to the contribution of the work. Also the color scheme was used for relations (red: less than, green: greater than and purple: roughly equal). Can the color scheme instead be used to show disagreement between templates i.e. to convey the information in L284-292?

**Questions:**

- The differentiation between pairwise instance scoring and model ordering is not clear. Authors should spend some time discussing the differences. Concretely, from L402-403, if a metric is a "good indicator on side-by-side comparison" (pair-wise instance scoring) then wouldn't it automatically be a good one to predict the "model ordering"? Do you mean pairwise instance scoring gives a partial ordering but model ordering should give a total order between models? How would a metric give a total order without comparing all models exhaustively? Please elaborate on the differences to highlight the importance of "model ordering" when a metric is already a good "pairwise instance" scorer.

---

> ### Author Response · Authors · 2024-11-15
> **Response to Reviewer GY7M**
>
> We thank the reviewer for their time and appreciate that they valued the rigorous and comprehensive evaluation. We respond to their questions below:
>
> **RQ1: “The paper requires multiple passes to assimilate...The terminology used is not intuitive and could be introduced earlier. Fig 2 is hard to read due to too many rows and the color scheme”**
>
> *Response:* We have updated the paper following your suggestions:
> 1. We have added a sentence “A skill refers to a generation challenge, such as text rendering or generating different colors and shapes and a sub-skill refers to sub-challenges (e.g.~generating longer text or Gibberish).” to the intro. And we have updated Table 8 in the appendix.
> 2. We have also updated Fig 2 to show only the results for Gecko(S) and Gecko(R). We included in Section D.2 of the Appendix (Figure 13) the complete figure showing results with all prompt sets. We also modified this figure to reflect the reviewer’s suggestion regarding the color code. Colors now represent whether all templates agree or not in the pairwise comparison. For Fig. 2 we kept the original color code as comparing templates is no longer necessary when reading this figure.
>
> **RQ2: Confusion around the difference between pairwise and model ordering.**
>
> *Response*: The difference between pairwise instance scoring and model ordering is the following.
>
> Pairwise instance scoring compares two metrics given *a pair of generations for a prompt*. Assume we have two generations $A(p), B(p)$ for the prompt $p$ for Model A and Model B. We also have a human preference (e.g. Model A $>$ Model B), and a metric $m$ which gives a score (e.g. $m(A(p), p)$) for a generation. Pairwise instance scoring would evaluate whether the relationship between the scores (i.e. $m(A(p), p) > m(B(p), p)$) matches that of the human rating. If the relationship matches, then we say $m$ is correct for that example, else it is incorrect.  To get an average accuracy for a given metric within a dataset we count the number of examples for which $m$ predicts the correct relationship for all prompts / model pairs and divide by the number of comparisons.
>
> For model ordering, instead we compare *two metrics over a set of prompts and corresponding generations*. We average the scores for a metric across all prompts to get $m_{avg}(A)= \sum_p {m(A(p), p)} / |P|$. We then evaluate (for all model pairs) how often the model ordering from human eval matches that obtained when comparing a given model pair: e.g. $m_{avg}(A)$ vs $m_{avg}(B)$ for Model A and Model B.
>
> Now, why these different evaluation procedures can give different results for a metric is subtle. Consider the following toy setting. We have two models: A and B, and 3 prompts. For prompt $p_1$, Model A is clearly much better than Model B, which is terrible, but for prompts $p_2,p_3$, Model A is ever so slightly worse but both are reasonable. Intuitively, Model A is better than Model B on this prompt set.
> - *Example 1*: Now imagine we’ve a metric that can do a good job of doing pairwise instance scoring but is not well calibrated across prompts (e.g. a score does not indicate an overall notion of alignment – 0.7 for one prompt could correspond to a poor generation but a score of 0.5 on another denotes a high quality generation). So in this toy example, that metric gives the following scores for Model A: ${p_1=0.1,p_2=0.1,p_3=0.1}$ and for Model B: ${p_1=0.05,p_2=0.8,p_3=0.8}$. The comparisons are all right, but the average score for Model A is $0.1$ and Model B is $0.55$, which does not match the intuition that Model A is actually better than Model B.
> - *Example 2*: We can conversely have a metric that is well calibrated across prompts but not able to reliably pick up on subtle differences. Such a metric could have the following scores for Model A: ${p_1=0.8,p_2=0.7,p_3=0.72}$ whereas Model B: ${p_1=0.1,p_2=0.69,p_3=0.71}$. While the model ordering is correct, the pairwise comparisons for prompts $p_2,p_3$ are not.
>
> These examples demonstrate how a metric can be good at pairwise comparison (e.g. a metric like CLIP) but be poor at model ordering, because a metric can give scores that are not well calibrated across prompts. Similarly, a metric can be good at model ordering but bad at pairwise instance scoring because it does not capture subtle differences.

---

> > ### Comment · Reviewer_GY7M · 2024-11-23
> > **Response to author comments**
> >
> > I appreciate the authors making changes following suggestions.
> > Thank you for explaining the differences between pairwise and model ordering. This wasn't clear at all from the text. Can you include a shorter version of the example in the paper (main or appendix) so that readers can properly understand what the authors mean? I like the example a lot, it gets to the point instantly.
> > I vote to keep my rating.

---

> > > ### Author Response · Authors · 2024-11-25
> > > **Response to Reviewer GY7M**
> > >
> > > We thank the reviewer for their suggestions. We have included the description and examples in the appendix (F.1) and have pointed the reader to these examples in Sec 6 (“App. F.1 gives a thorough description of each task and intuitive examples for how they differ”).

---

### Official Review · Reviewer_kjYy · 2024-11-07

**Soundness:** 3
**Presentation:** 3
**Contribution:** 2
**Rating:** 6
**Confidence:** 4

**Summary:**

This work introduces a comprehensive evaluation suite with over 100,000 annotations and four human annotation templates to assess T2I model capabilities across diverse conditions. Key contributions include a curated prompt set (Gecko2K), a statistical method for comparing models, and a framework for systematic metric evaluation across three tasks. The results suggest that metrics vary in effectiveness across settings, and to overcome these issues the authors propose an interpretable auto-eval metric that consistently correlates better with human ratings.

**Strengths:**

- Proper comparison with previous works
- Through evaluation of the proposed metric and the dataset
- The authors have proposed a new benchmark dataset to highlight the issue of evaluation across multiple templates
- Furthermore, the authors propose a new metric for evaluating QA/VQA models
- The paper is well-written and easy to follow

**Weaknesses:**

- My primary concern is with the validity of the metric proposed:
  - A central aspect of the proposed metric is its use of NLI to detect hallucinations, making the metric reliant on the accuracy of the NLI model. Errors in the NLI model could undermine confidence in the reliability of this metric.
  - The proposed alignment metric in Equation 2 relies entirely on the negative log-likelihood (NLL) scores generated by the visual question-answering (VQA) model to evaluate alignment. NLL scores measure how well the model predicts a correct answer given an image and a question prompt; lower scores indicate higher confidence in the answer. However, this approach assumes that the VQA model’s confidence is accurate and does not account for situations where the model might be overly confident (assigning low NLL scores even when it’s wrong) or under-confident (assigning high NLL scores even when it’s correct).
If the model is over-confident, it might produce low NLL scores for incorrect or poorly aligned answers, leading to artificially high alignment scores. Conversely, if the model is under-confident, it might assign high NLL scores even for well-aligned responses, causing the alignment score to underestimate the actual alignment. Because Equation 2 relies solely on these NLL scores without adjustments, it doesn’t account for these potential biases in model confidence, which could compromise the accuracy and reliability of the alignment metric.

**Questions:**

- Since the proposed metric depends heavily on the accuracy of the NLI model to detect hallucinations, how do you address potential errors from the NLI model that could impact the reliability of your metric? Have you considered any methods to mitigate this dependency?
- The alignment metric in Equation 2 assumes that the VQA model’s confidence, as measured by NLL scores, is accurate. How do you account for cases where the VQA model might be overly confident in incorrect answers or under-confident incorrect ones?
- Given that low NLL scores could indicate either true alignment or over-confidence in incorrect answers, and high NLL scores could indicate either misalignment or under-confidence incorrect answers, have you considered additional adjustments or alternative metrics to address this bias?
- Since the proposed metric doesn’t currently account for potential biases in the VQA model’s confidence, how might this impact the interpretability and reliability of alignment scores? Would adding a calibration step improve the metric’s robustness?

Post-rebuttal: After reviewing the authors' responses, I am increasing my score from 5 to 6.

---

> ### Author Response · Authors · 2024-11-15
> **Response (1/2)**
>
> We thank the reviewer for their time and appreciate that they valued the thorough evaluation.
>
> As a point of clarification, we do not expect the underlying models (VQA, NLI) to be flawless but show that, despite their flaws, our metric is performant and reliable. To validate our metric we use a comprehensive and rigorous evaluation setup showing our performant alignment metric is accurate, obtaining 72/79% agreement with human preference and on average 0.53 correlation – Table 4. Our results and ablations show:
> 1) models nowadays are advanced enough to create robust, performant metrics (which perform significantly better than existing metrics such as CLIP) – Table 4.
> 2) with further progress made in the LLM and VQA space, this metric will only become more accurate – Table 4
>
> Note that it is outside the scope of this work to retrain or recalibrate those underlying models. Below, we discuss how we address some of the potential limitations of the underlying components as asked by the reviewer.
>
> **RQ1: “A central aspect of the proposed metric is its use of NLI to detect hallucinations, making the metric reliant on the accuracy of the NLI model. Errors in the NLI model could undermine confidence in the reliability of this metric.” … “Since the proposed metric depends heavily on the accuracy of the NLI model to detect hallucinations, how do you address potential errors from the NLI model that could impact the reliability of your metric? Have you considered any methods to mitigate this dependency?”**
>
> *Response:* There are actually two effects that help minimise hallucination: (1) the reliability of the NLI metric; and (2) the quality of the LLM model in not hallucinating in the first place. As regards (1), we are using the NLI metric in a simple setting – given the prompt and the question, is the question grounded in the prompt – so it should be more reliable than in more complex settings. To verify this, we manually evaluated a random selection of 1.8K QA pairs evenly chosen from Gecko(R) and Gecko(S) and found that the NLI metric is ~93% accurate on the LLM generated set of QA pairs. This corroborates how in Table 5, we see that adding the NLI filtering method improves performance. We also compare how often the NLI metric removes questions for an older model (PALM-2) versus a newer model (Gemini) and find that in the first case, we remove 13% of the questions and in the second 2%. As a result, we find that as LLM models improve, the importance of this step will decrease.
>
> **RQ2: The alignment metric in Equation 2 assumes that the VQA model’s confidence, as measured by NLL scores, is accurate. How do you account for cases where the VQA model might be overly confident in incorrect answers or under-confident in correct ones?**
>
> *Response:* The performance of our metric does depend on the accuracy of the VQA module and VQA errors could degrade the overall accuracy of the metric. However, by breaking the evaluation down into multiple questions, we are more robust to over/underconfidence of answers to a single question. We also validate our use of the VQA module using our human eval data. If the VQA model were inaccurate and overconfident or underconfident as suggested, the metric would perform poorly as it would not align with human annotation. Given that performance is high, obtaining 72-79% agreement with human preference and on average 0.53 correlation, we can have confidence that the VQA model is accurate and the scores are reasonable.
> The finding that the VQA component is accurate and scores meaningful is corroborated by other work ([1], table 4), which has specifically evaluated the quality of the VQA model itself and it shows that PALI (which we use) and even weaker backbones than what we use achieve high accuracy.

---

> ### Author Response · Authors · 2024-11-15
> **Response (2/2)**
>
> **RQ3: Given that low NLL scores could indicate either true alignment or over-confidence in incorrect answers, and high NLL scores could indicate either misalignment or under-confidence incorrect answers, have you considered additional adjustments or alternative metrics to address this bias? Since the proposed metric doesn’t currently account for potential biases in the VQA model’s confidence, how might this impact the interpretability and reliability of alignment scores? Would adding a calibration step improve the metric’s robustness?**
>
> *Response:* As discussed in the previous response, the VQA models we use are generally accurate. Moreover, for each model, where reasonable – e.g. PaLI-17b, we used the versions with the largest language components which have been shown to yield better calibrated VQA models [2]. We further refer the reviewer to Table 5 from our manuscript, where we showed that the score normalization (as opposed to just using the raw answer) yields a metric that is better correlated with human judgement across both Gecko(R) and Gecko(S) and all evaluation templates. This finding leads us to hypothesise that our metric would improve in case a better calibrated VQA model was available, but training a better calibrated VQA model is beyond the scope of our contributions, as we focus on treating the underlying models as ‘black boxes’ in the line of other QA/VQA work (e.g. TIFA [4], DSG [1]).
>
> We note that if there were a strong bias, e.g. the model always outputs ‘yes’, then we would perform very poorly in our evaluations at side by side preference, as the score would be independent of the visual input. Seeing as scores here are high (and higher than other baselines), obtaining high agreement with human preference (Table 4), we can have confidence that such a bias is minimal. We additionally did a sanity check by exploring how our VQA models respond to ‘blind’ questions. Given no image and a question, will the VQA model always output a given answer. Here we focus on yes/no questions as the yes bias has been noted in older models [3] and the majority of our questions are binary ones due to the few-shot prompt. However, we find that with Gemini, we obtain only 20% ‘yes’ answers in this setting and 80% ‘no’ indicating that the model does not have a strong ‘yes’ bias. We also note that if a model is biased, it is equally biased for any input, which means that a relative comparison is still valid. Again, given the high performance of the metric we can infer that such a problematic bias does not exist and so we can have confidence in the utility and validity of the metric.
>
> [1] Davidsonian Scene Graph: Improving Reliability in Fine-grained Evaluation for Text-to-Image Generation. Cho et al. ICLR 2024.
>
> [2] Vasily Kostumov, Bulat Nutfullin, Oleg Pilipenko, Eugene Ilyushin. Uncertainty-Aware Evaluation for Vision-Language Models, 2024.
>
> [3] Aishwarya Agrawal, Dhruv Batra, Devi Parikh, and Aniruddha Kembhavi. Don’t just assume; look and answer: Overcoming priors for visual question answering. CVPR 2018
>
> [4] TIFA: Accurate and Interpretable Text-to-Image Faithfulness Evaluation with Question Answering. Hu et al. ICCV 2023.

---

> > ### Comment · Reviewer_kjYy · 2024-11-25
> >
> > Thank you to the authors for their detailed response. Most of my concerns have been addressed. However, the issue of hallucination remains and appears somewhat unavoidable, making it essential to include a proper discussion. The same applies to the inaccuracies of the VQA model. Since these issues cannot be fully resolved, it is crucial to incorporate a thorough discussion, ablation studies, and empirical analysis. I recommend that the authors include this discussion in the manuscript, at least in the supplementary section.
> >
> > While my main concerns have been addressed, I want to clarify the reason for my relatively low score. The primary contribution of the paper is a new dataset, which I believe is more suited for a dataset and benchmark track. This is a personal opinion, and my score is not reflective of the quality or contribution of the work itself. Therefore, I would like to flag this for the Area Chair’s attention, and they are welcome to disregard my score if deemed appropriate. Accordingly, I am increasing my score to 6.

---

> > > ### Author Response · Authors · 2024-11-25
> > > **Response to reviewer kjYy**
> > >
> > > We thank the reviewer for their suggestions and have added the suggested discussion to C.3 in the appendix.
> > >
> > > We appreciate the reviewer’s concern regarding whether our paper is “more suited for a dataset and benchmark track”. We first highlight that the contributions within our work include a thorough analysis of human evaluation templates, as well as a new state-of-the-art metric for measuring text-to-image alignment, therefore, going much beyond only introducing a new benchmark. Additionally, we remark that, in the 2025 ICLR call for papers, “Datasets and benchmarks” are explicitly mentioned in the list of relevant topics (c.f. the 14th bullet point on the [call for papers page](https://iclr.cc/Conferences/2025/CallForPapers)) and papers accepted at recent editions of ICLR included contributions focused on proposing new datasets and benchmarks. We present a non-exhaustive list of such accepted papers:
> > > - MIntRec2.0: A Large-scale Benchmark Dataset for Multimodal Intent Recognition and Out-of-scope Detection in Conversations, ICLR 2024
> > > - SWE-Bench: Can Language Models Resolve Real-World GitHub Issues?, ICLR 2024 (oral)
> > > - MEDFAIR: Benchmarking Fairness for Medical Imaging, ICLR 2023 (notable-top-25%)
> > > - GAIA: a benchmark for General AI Assistants, ICLR 2024
> > >
> > > In light of that, we strongly believe that our work is well-suited for the ICLR community and we kindly ask the reviewer to consider these points when evaluating our work.

---

### Public Comment · ~Olivia_Wiles1 · 2025-02-18
**Updated paper title.**

We have updated the paper title to better highlight the three main contributions in the work. We have made only a handful of superficial changes within the text of the paper.

---

### Meta-Review · Area_Chair_3XKi · 2024-12-18

**Metareview:**

(a) The paper introduces Gecko2K, a robust benchmark and auto-evaluation metric for text-to-image (T2I) alignment, rigorously validating evaluation tasks, human templates, and metrics. Key findings show significant metric variability across settings and demonstrate superior human correlation with the proposed method.
(b) Strengths: Comprehensive analysis, novel benchmark, new auto-eval metric, and practical insights for T2I research.
(c) Weaknesses: Reliance on VQA/NLI models, limited scale (4 models, 2K prompts), and clarity issues in terminology and visuals.
(d) Decision: Accept. The paper provides strong contributions, sound evaluations, and actionable findings, advancing T2I alignment research despite minor limitations.

**Additional Comments On Reviewer Discussion:**

Reviewers raised concerns about the clarity of terminology, potential biases in the VQA/NLI components, and dataset scalability. The authors clarified terms, improved figures, provided ablations, and added discussions on model reliability. These efforts addressed reviewer concerns, showing the paper’s robustness and generalizability, supporting its acceptance decision despite minor weaknesses.

---

### Decision · Program_Chairs · 2025-01-22

Accept (Spotlight)